# Fisetin inhibits *Salmonella* Typhimurium type III secretion system regulator HilD and reduces pathology *in vivo*

Siqi Li,[1] Hongtao Liu,[1] Jingyan Shu,[1] Quanshun Li,[2] Yuan Liu,[3] Haihua Feng,[1] Jianfeng Wang,[1] Xuming Deng,[1] Yong Zhang,[1] Zhimin Guo,[1] Jiazhang Qiu[1]

**ABSTRACT** *Salmonella enterica* is an important zoonotic intracellular bacterial pathogen that is capable of causing infections ranging from localized gastroenteritis to fatal systemic infection in humans and food-producing animals. The increasing antibiotic resistance in *Salmonella* isolates, especially the emergence of MDR and newer XDR strains, has compromised the efficacy of conventional antimicrobial therapy for *Salmonella* infections. Hence, it is desirable to develop alternative therapeutic means to tackle the antimicrobial resistance crisis. In this study, we screened plant-derived compounds to identify inhibitors of *Salmonella* invasion of host cells. These efforts identified fisetin as a possible protector against infection. Further mechanistic studies revealed that fisetin suppressed the function of type III secretion system 1 (T3SS-1), the virulence determinant critical for *Salmonella* invasion. Fisetin appears to interfere with the interaction between HilD and its relevant promoters, thereby decreasing the transcription of *hilA*, the central transcriptional regulator that functions to activate the expression of T3SS-1 effector proteins and structural elements. In addition, administration of fisetin in the *Salmonella* murine infection model resulted in reduced bacterial colonization, alleviation of histopathological destruction, and decreased proinflammatory cytokine levels. Taken together, our study establishes that the natural compound fisetin can be used as a lead compound for the development of anti-*Salmonella* drugs targeting T3SS-1.

**IMPORTANCE** Salmonella spp. remains a major worldwide health concern that causes significant morbidity and mortality in both humans and animals. The spread of antimicrobial resistant strains has declined the efficacy of conventional chemotherapy. Thus, novel anti-infection drugs or strategies are needed. Anti-virulence strategy represents one of the promising means for the treatment of bacterial infections. In this study, we found that the natural compound fisetin could inhibit Salmonella invasion of host cells by targeting SPI-1 regulation. Fisetin treatment impaired the interaction of the regulatory protein HilD with the promoters of its target genes, thereby suppressing the expression of T3SS-1 effectors as well as structural proteins. Moreover, fisetin treatment could reduce pathology in the Salmonella murine infection model. Collectively, our results suggest that fisetin may serve as a promising lead compound for the development of anti-Salmonella drugs.

**KEYWORDS** *Salmonella*, fisetin, T3SS-1 inhibitor, anti-infection, anti-virulence

The increasing antimicrobial resistance (AMR) in diverse bacterial pathogens has raised tremendous public health concerns. In particular, the emergence and rapid dissemination of multidrug-resistant (MDR) strains carrying mobile MCR (1), NDM (2), and TetX (4/5) (3, 4) genes has greatly threatened the therapeutic efficacy of last-resort antibiotics (colistin, carbapenems, and tigecycline). In 2019, there were an estimated

Address correspondence to Jiazhang Qiu, qiujz@jlu.edu.cn, Zhimin Guo, amily@jlu.edu.cn, or Yong Zhang, zhangybh@jlu.edu.cn.

Siqi Li and Hongtao Liu contributed equally to this article. Author order was determined by the corresponding authors after negotiation.

The authors declare no conflict of interest.

See the funding table on p. 15.

4.95 million bacterial AMR-associated deaths, including 1.27 million deaths directly caused by bacterial AMR (5). By 2050, it was estimated that approximately 10 million people will die of AMR every year, which could cost the global economy up to 100 trillion dollars (6). Consequently, there is a pressing need for the development of new antibiotics or therapeutic strategies to tackle this crisis.

*Salmonella enterica* is a facultative intracellular bacterial pathogen that is capable of causing localized or systemic infections in both humans and animals, resulting in huge economic costs and public health concerns worldwide (7). Approximately 1.3 billion clinical cases of gastroenteritis and 16 million cases of typhoid fever are documented annually worldwide, which in total cause 3 million *Salmonella* infection-related annual deaths (8). In particular, gastroenteritis remains one of the primary causes of morbidity and mortality in children younger than 5 years (9). The development of vaccines to prevent *Salmonella* infection appears to be unsuccessful due to their poor cross-protection against distinct serovars (10). Therefore, antibiotics are the preferred treatment for diseases caused by *Salmonella* in humans and animals (11). Unfortunately, the widespread misuse of antibiotics has greatly promoted the rapid emergence and development of antibiotic resistance. Approximately 60% of *Salmonella* clinical isolates have developed resistance against first-line antibiotics (12). In particular, the emergence of extensive drug-resistant (XDR) *Salmonella* strains has raised global public health concerns (13). Consequently, it is highly necessary to develop novel anti-infection drugs or therapeutic strategies to control *Salmonella* infections.

One of the critical properties of *Salmonella* pathogenesis is its ability to invade and replicate within host cells (14). To achieve this, pathogenic *Salmonella* is equipped with multiple virulence determinants, among which type III secretion systems 1 and 2 (T3SS-1 and T3SS-2) encoded by *Salmonella* pathogenicity islands 1 and 2 are the most important and well characterized in *Salmonella* infections (11, 15). T3SS-1 and T3SS-2 are molecular syringes that directly translocate a number of effector proteins into infected host cells (16). Once within host cells, T3SS-1 effectors trigger cytoskeletal rearrangements and membrane ruffling, which are critical for bacterial engulfment and the proinflammatory response (17). The expression of T3SS-1 genes is tightly controlled by complex regulatory networks, among which the OmpR/ToxR family protein HilA is the central transcriptional regulator (18). HilA can activate all the operons coding for the functional T3SS-1, including the *prg/org* and *inv/spa* operons (11). The transcription of *hilA* is regulated by three homologous proteins, RtsA, HilC, and HilD, which form a complicated feed-forward regulatory loop (11). Of these, HilD serves as a switch to integrate environmental signals and activate the expression of *hilC*, *rtsA,* and *hilA* (11). Owing to their essential roles in pathogenesis, *Salmonella* T3SSs have been recognized as promising targets for novel anti-infection strategies.

Fisetin is a flavonoid compound that is naturally found in various fruits and vegetables and is also found in some trees belonging to the Fabaceae and Anacardiaceae families (19). Fisetin has been proven to possess diverse pharmacological properties, such as anticoagulant (20), antioxidant (21), antitumor (22), and neuroprotective (23) activities. In this study, based on the *Salmonella* invasion assay, we initiated drug screening to search for invasion inhibitors. From a library containing 550 natural compounds, we found that treatment of *Salmonella* with fisetin could significantly decrease its invasion of host cells. Further mechanistic studies revealed that fisetin could interfere with the binding of HilD, the major regulator of T3SS-1, with its relative promoters, thus reducing the production of T3SS-1 effectors and structural proteins. In addition, fisetin treatment showed reduced organ bacterial colonization, alleviated histopathological destruction, and decreased cytokine levels in a murine *Salmonella* infection model. Our data established that fisetin could be used as a lead compound for anti-*Salmonella* drugs targeting T3SS-1. Moreover, fisetin may be applied in combination with existing therapies (e.g., antibiotics) for the treatment of *Salmonella* infections.

## RESULTS

### Fisetin inhibits *Salmonella* invasion of HeLa cells

In this study, we screened 550 natural compounds to identify active small molecules that can inhibit *Salmonella* invasion of HeLa cells *via* the gentamicin protection assay (Fig. 1A). These efforts have allowed us to identify fisetin, a plant-derived flavonoid compound, as a positive hit. Incubation of *S.* Typhimurium with increasing concentrations of fisetin dose-dependently decreased the uptake of bacteria by the host cells (Fig. 1B), as determined by counting the colony forming units (CFUs) on the plates. Similar observations were obtained by immunostaining the infected cells with *Salmonella*-specific antibodies (Fig. 1C). In both assays, as a negative control, HeLa cells challenged with *S.* Typhimurium Δ*invA*, a T3SS-1-deficient strain, showed little bacterial invasion (Fig. 1B and C). In addition, fisetin treatment also showed an inhibitory effect on *S.* Enteritidis invasion of HeLa cells (Fig. S1A and B). The minimal inhibitory concentrations (MICs) of fisetin against *S.* Typhimurium and *S.* Enteritidis were greater than 1024 µg/mL (Fig. 1D; Fig. S1C). In addition, incubation of both strains with 64 µg/mL fisetin did not affect the bacterial growth rate, as determined by measuring the optical density of the culture at 600 nm (Fig. 1E; Fig. S1D). Taken together, our data indicate that non-antibacterial fisetin inhibits *Salmonella* invasion.

### Fisetin decreases the secretion levels of *Salmonella* T3SS-1

According to the abovementioned observations as well as the essential roles of T3SS-1 in *Salmonella* entry into nonphagocytic cells (17), we next investigated the effect of fisetin on the function of T3SS-1. To this end, we first applied the beta-lactamase (TEM)

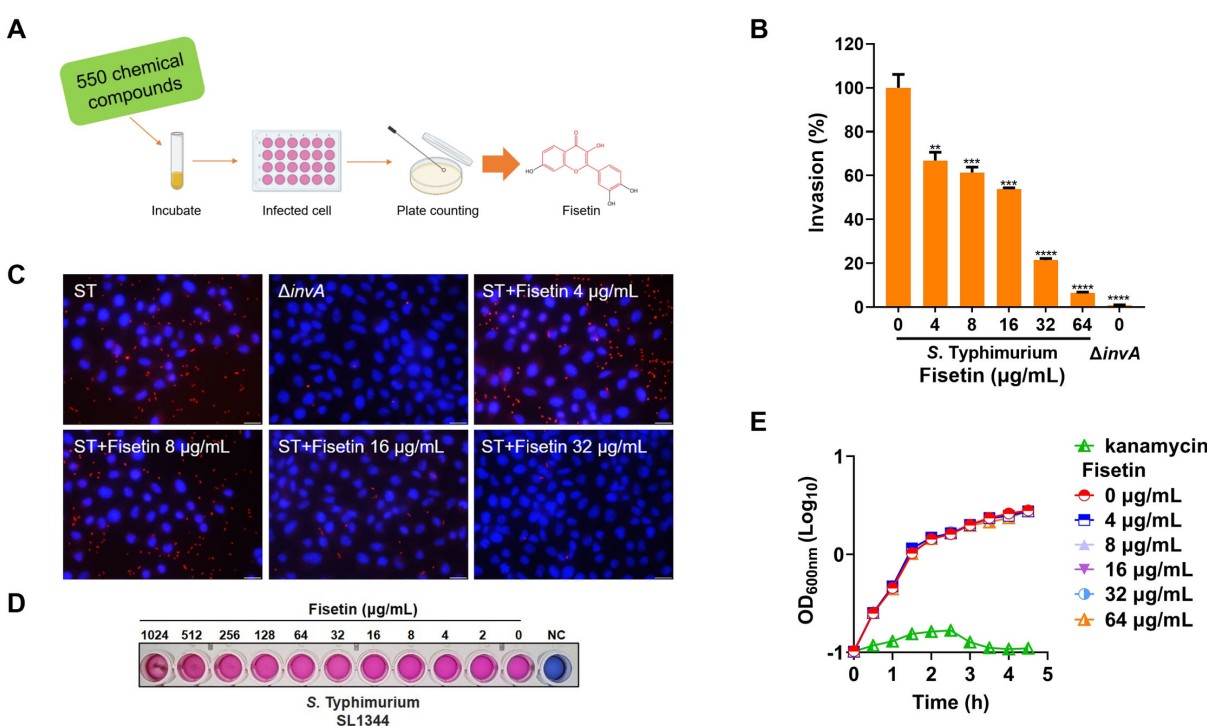

**FIG 1** Fisetin inhibits *S.* Typhimurium invasion of HeLa cells. (A) Schematic diagram of screening for *S.* Typhimurium invasion inhibitors from natural compounds. (B) Growth of *S.* Typhimurium with fisetin suppresses bacterial invasion of host cells. Bacteria cultured with increasing concentrations of fisetin (0–64 µg/mL) were used to infect HeLa cells. Percent invasion of the fisetin-treated samples was standardized to the untreated control, which served as 100%. (C) Immunofluorescence of intracellular *S.* Typhimurium. Bar = 20 µm. The *S.* Typhimurium Δ*invA* that barely invaded epithelial cells was included as a negative control. (D) The MIC of fisetin against *S.* Typhimurium SL1344 was determined by the broth microdilution method. NC indicates negative control. (E) *In vitro* growth curve of *S.* Typhimurium in the presence of various concentrations of fisetin. Kanamycin at 4 µg/mL was used as a positive control. Data presented in panel B are the mean ± SD of three independent experiments. Panels C, D, and E are representative of three independent experiments. ****$P < 0.0001$; ***$P < 0.001$; **$P < 0.01$.

system to visualize the translocation of *Salmonella* T3SS-1 effector proteins into infected cells. HeLa cells seeded on 24-well plates were infected with *S.* Typhimurium SL1344 (pSipA-TEM) for 2 h and loaded with the TEM substrate CCF4-AM. Then, the translocation of SipA-TEM was observed under a fluorescence microscope. Without fisetin treatment, most of the infected cells exhibited blue fluorescence, indicating efficient effector translocation (Fig. 2A and B). However, incubation of *S.* Typhimurium with fisetin led to a dose-dependent reduction in protein delivery into the host cytosol, as revealed by the decrease in the percentage of blue fluorescence cells and increase in the percentage of green fluorescence cells (Fig. 2A and B). As a negative control, the expression of SipA-TEM in *S.* Typhimurium Δ*invA* failed to deliver the fusion proteins into the infected cells (Fig. 2A and B).

To further explore the role of fisetin on T3SS-1, we measured the levels of endogenous T3SS-1 substrates that were secreted into the culture supernatants. As shown in Fig. 2C, the secretion levels of T3SS-1 effectors, including SipA, SipB, and SipC, were decreased upon treatment with fisetin. Moreover, we also analyzed the TCA precipitates by western blot using SipA- and Flag-specific antibodies (Fig. 2D). The data showed that 4 µg/mL fisetin resulted in significantly reduced secretion of SipA and SipB, whereas effector levels were almost undetectable in the samples receiving 8 µg/mL fisetin (Fig. 2D). By contrast, the secretion of flagellin C (FliC) was not affected by either concentration of fisetin (Fig. 2D). In addition, a similar reduction in T3SS-1 effector secretion was observed in fisetin-treated *S.* Enteritidis (Fig. S2A and S2B). Together, these data suggest that fisetin could inhibit the secretion of *Salmonella* T3SS-1.

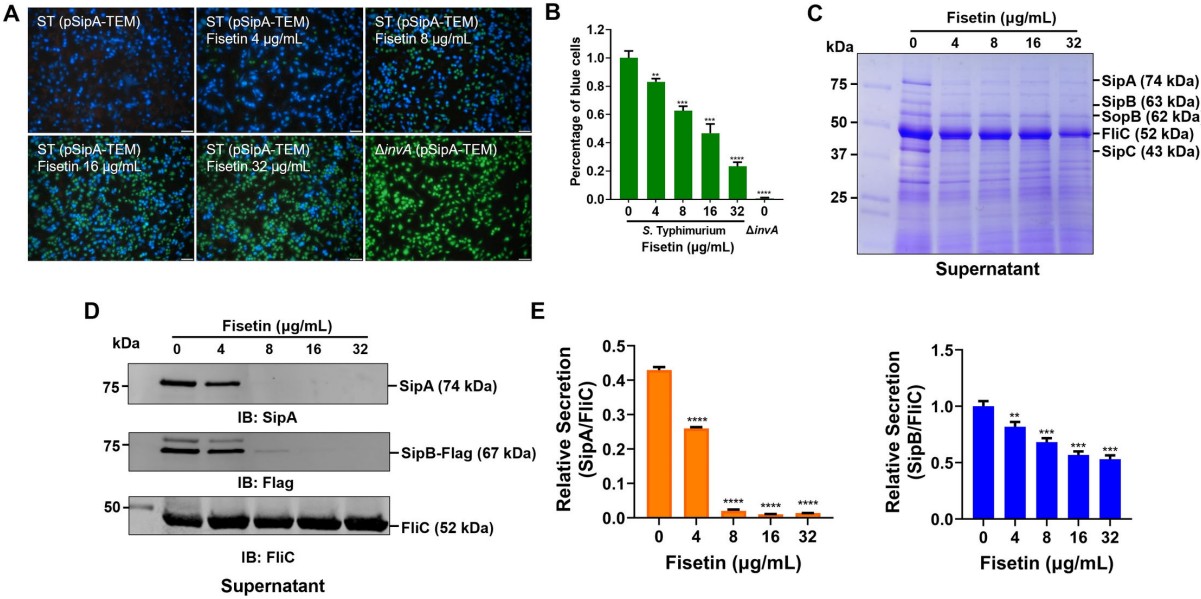

**FIG 2** Fisetin treatment decreases the secretion of *S.* Typhimurium T3SS-1. (A) Translocation of SipA-TEM into HeLa cells. *S.* Typhimurium expressing SipA-TEM was either untreated or incubated with increasing concentrations of fisetin. Bacteria were added to the HeLa cell culture and infected for 2 h, and SipA-TEM translocation was visualized under a fluorescence microscope after loading with CCF4. Bar, 50 µm. (B) Percent of SipA-TEM delivery into the host cells was determined by calculating the blue fluorescence cells out of the total cells. *S.* Typhimurium Δ*invA*, which is deficient in functional T3SS-1, was used as a negative control. (C-D) The secretion levels of T3SS-1 effectors in the *S.* Typhimurium culture supernatants. After incubation with fisetin, the bacterial culture supernatants were collected and subjected to TCA precipitation. The amounts of effectors were evaluated by SDS−PAGE followed by either Coomassie brilliant blue (CBB) staining (C) or western blot analysis with specific antibodies against SipA, Flag, and FliC (D). FliC was probed as a loading control. (E) Relative protein levels were determined by ImageJ to calculate the density ratios of SipA or SipB against FliC. Data in Panels A, C, and D are one representative of three independent assays, while data in B and E are the mean ± SD of three independent experiments. ****$P < 0.0001$, $P < 0.0001$; ***$P < 0.001$; **$P < 0.01$.

## Fisetin suppresses the expression and transcription of T3SS-1 effectors

To further investigate the exact mechanism of decreased secretion, we detected the expression levels of effector proteins in *Salmonella* cultures treated with increasing concentrations of fisetin. Remarkably, the production of plasmid-derived SipA-TEM was significantly decreased in samples cultured with fisetin, as measured by western blot using an anti-TEM antibody (Fig. 3A). In addition, the expression of endogenous SipA and SipB was also remarkably inhibited upon fisetin treatment (Fig. 3B).

Next, we tested whether fisetin affects the transcription of *Salmonella* T3SS-1 effector-encoding genes. After treatment of *Salmonella* with fisetin, the mRNA levels of SipA, SipB, and SipC were measured by real-time RT-PCR. Similar to the protein expression levels, fisetin treatment significantly suppressed the transcription of genes coding for T3SS-1 effector proteins (Fig. 3C). Taken together, our data indicate that fisetin inhibits *Salmonella* T3SS-1 at the transcriptional level, thereby reducing the production and subsequent secretion of T3SS-1 effector proteins.

## Inhibition of the T3SS by fisetin is dependent on its suppression of HilA

The abovementioned observations prompted us to examine whether fisetin affects the expression of any of the T3SS-1 regulatory genes (Fig. 4A). To this end, we first measured the transcriptional levels of *hilA*, *hilC*, *hilD*, and *rtsA* in *S.* Typhimurium SL1344 incubated with fisetin. Remarkably, fisetin suppressed the transcription of these genes in a dose-dependent manner (Fig. 4B). When fisetin was used at 64 µg/mL, the transcription of *hilA*, *hilC*, *hilD*, and *rtsA* was reduced by 1.9-, 2.8-, 1.8-, and 2.7-fold compared to the untreated control (Fig. 4B). In line with this, the production of HilA was significantly impaired in the presence of fisetin, as shown by western blot analysis using a HilA-specific antibody (Fig. 4C). In addition, we constructed a HilA overexpression plasmid and introduced it into *S.* Typhimurium SL1344. Following the treatment of *S.* Typhimurium

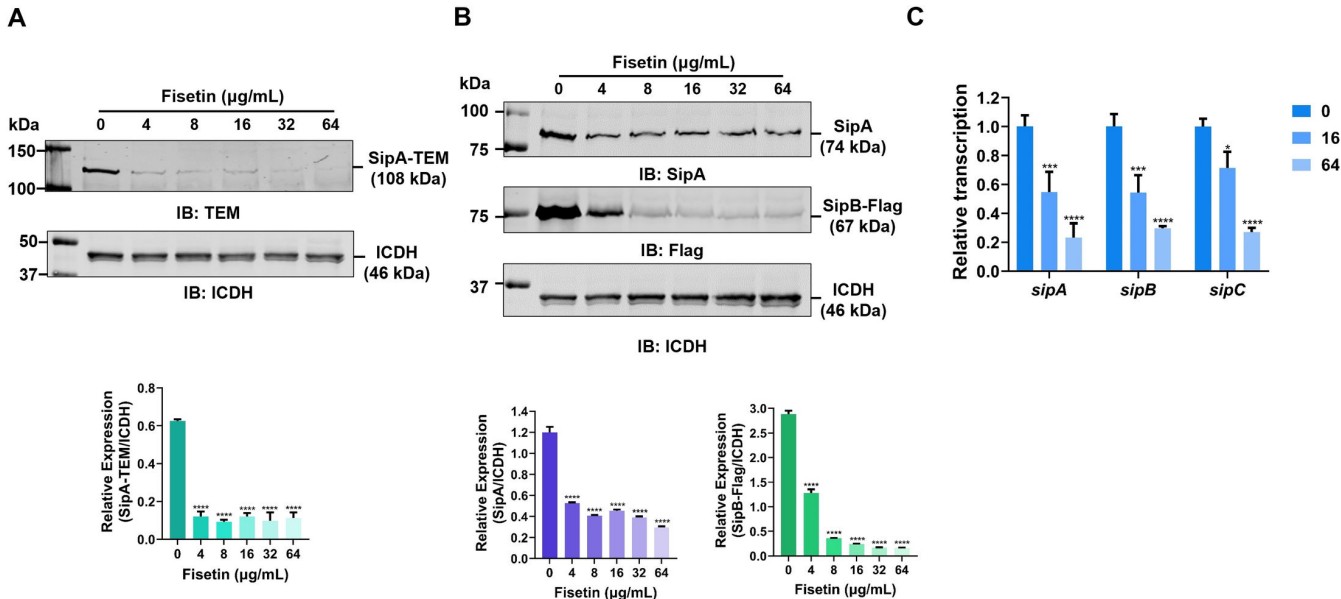

**FIG 3** Fisetin reduces the expression of *S.* Typhimurium T3SS-1 effectors. (A) Expression of SipA-TEM in *S.* Typhimurium (pSipA-TEM) treated with gradient doses of fisetin. Equal amounts of bacteria were pelleted and resuspended in 1× SDS sample buffer. SipA-TEM was detected by western blot after SDS–PAGE using an anti-TEM antibody. ICDH was probed as a loading control. Relative protein levels were determined by ImageJ to calculate the density ratios of SipA against ICDH. (B) Production of endogenous SipA and SipB in fisetin-treated bacteria. A similar procedure was used as described in (A) except that proteins were probed with anti-SipA and Flag antibodies. (C) Transcriptional levels of *S.* Typhimurium T3SS-1 effector coding genes. Bacteria were either untreated or cocultured with 16 µg/mL or 64 µg/mL fisetin. The transcription of *sipA*, *sipB*, and *sipC* was measured by qRT–PCR. ****$P < 0.0001$; ***$P < 0.001$; *$P < 0.05$. Data shown in Panel C are the mean ± SD of three independent experiments.

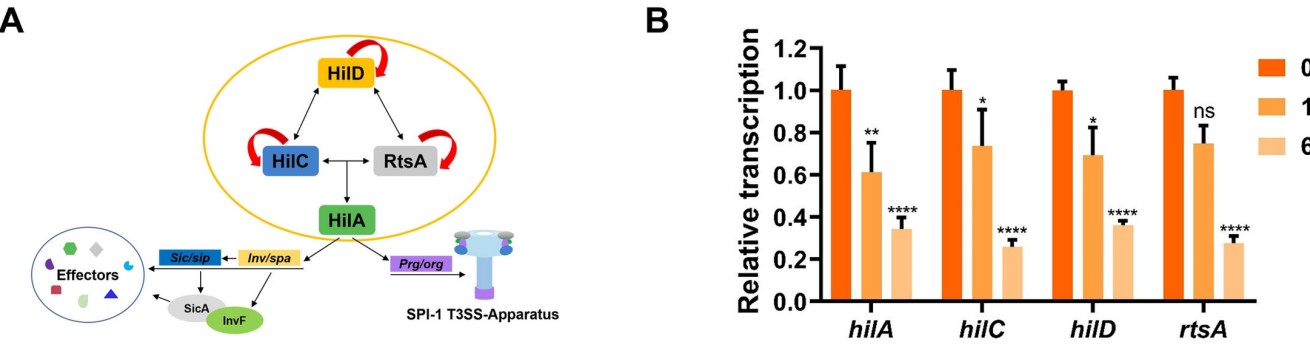

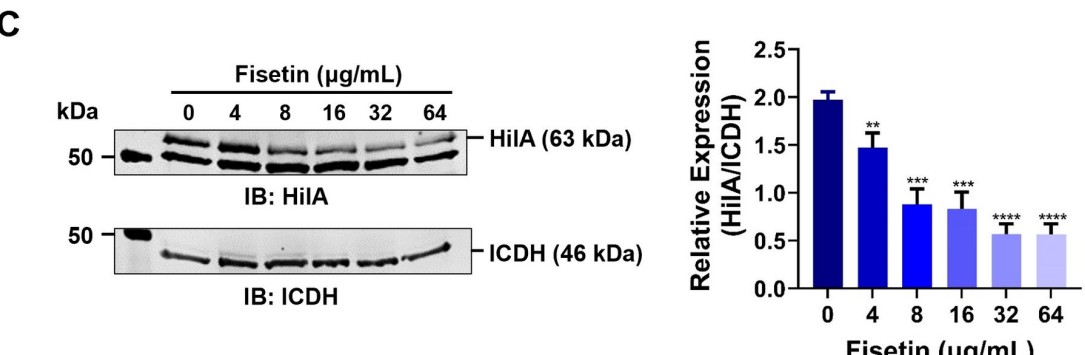

**FIG 4** Fisetin treatment suppresses the expression and transcription of the T3SS-1 regulator HilA. (A) Schematic diagram of the T3SS-1 regulatory network. (B) Relative transcription of *hilA*, *hilC*, *hilD*, and *rtsA*. *S.* Typhimurium was grown in the absence or presence of fisetin. Total RNA was isolated and further reverse transcribed into cDNA. The mRNA levels were determined by qRT–PCR using specific primers. Data shown in B are the mean ± SD of three independent experiments. (C) HilA production in *S.* Typhimurium upon treatment with increasing doses of fisetin. Protein samples separated by SDS–PAGE were further analyzed by western blotting using an anti-HilA antibody. ICDH was probed as a loading control. Relative repression of HilA was quantified by ImageJ and is shown in the right panel. ****$P < 0.0001$; ***$P < 0.001$; **$P < 0.01$; *$P < 0.05$.

SL1344 (pJL03-*hilA*) with increasing fisetin concentrations, the inhibition of SipA was restored upon the induction of plasmid-derived HilA (Fig. 5A and B).

In addition to activating the transcription of T3SS-1 effector-encoding genes, HilA can also regulate the expression of T3SS-1 structural proteins (24). Since fisetin treatment could decrease the level of HilA in *S.* Typhimurium, we speculated that the expression of the T3SS-1 structural gene might also be transcriptionally regulated by fisetin. To test this hypothesis, we first performed real-time RT-PCR to quantify the mRNA levels of *prgH*, *prgI*, *prgK*, and *invG* in *S.* Typhimurium after incubation with fisetin. Compared to the untreated control, the relative transcripts of *prgH*, *prgI*, *invG*, and *prgK* in *S.* Typhimurium challenged with 64 µg/mL fisetin were decreased by 2.6-, 10.1-, 24-, and 4.9-fold, respectively (Fig. 6A). Further western blot analysis using PrgH-specific antibodies also showed a dose-dependent reduction in PrgH production in fisetin-treated *S.* Typhimurium. Similarly, overproducing HilA in *S.* Typhimurium by an arabinose-inducible plasmid can block the inhibition of PrgH production by fisetin (Fig. 6B and C). Taken together, our data suggest that the fisetin-induced suppression of T3SS-1 is primarily mediated by lowering the production of HilA.

## Fisetin impairs the binding between HilD and promoter DNA sequences

Within the T3SS-1 regulatory circuit, HilD plays a predominant role in activating the transcription of *hilC*, *rtsA*, and *hilA* by directly binding to their promoters (11). HilD is also self-regulated by binding to its own promoter region (8). Based on the aforementioned results, we hypothesized that fisetin might disrupt the interaction between

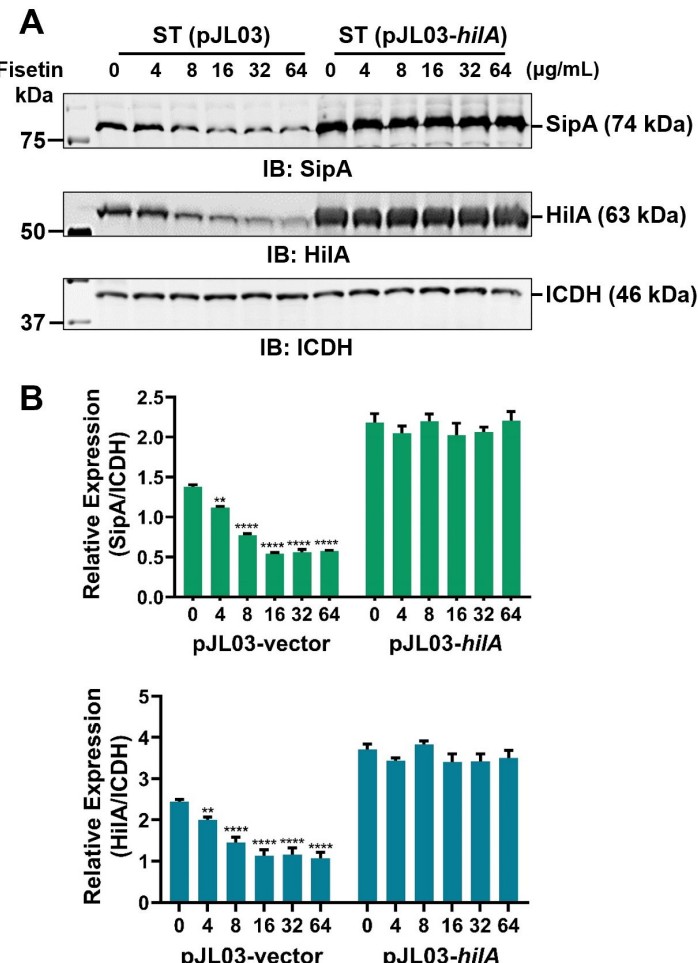

**FIG 5** Overexpression of HilA in *S.* Typhimurium counteracts fisetin-mediated inhibition of T3SS-1. (A) *S.* Typhimurium was transformed with the empty vector pJL03 or pJL03-*hilA,* which drives the arabinose-induced expression of HilA. After incubation of these strains with increasing doses of fisetin, the protein levels of SipA and HilA in the bacterial lysates were detected by western blot analysis using anti-SipA and anti-HilA antibodies. Anti-ICDH was used as a loading control. (B) Relative expression of SipA and HilA was determined by quantification of the blot density using ImageJ. Data are representative of three independent experiments. ****$P < 0.0001$; **$P < 0.01$.

HilD and its target promoters. To this end, we employed an electrophoretic mobility shift assay (EMSA) to test this hypothesis using recombinant His$_6$-HilD and relevant promoter probes. In the absence of fisetin, HilD efficiently interacted with the *hilA* and *hilD* promoters, as shown by the formation of the HilD-promoter complex. However, the binding of HilD with DNA probes was gradually impaired along with the addition of increasing dosages of fisetin to the *in vitro* reaction, which is evident by the increase in free DNA and decrease in the HilD-DNA complex visualized on the gel (Fig. 7A and B). Taken together, these results suggest that fisetin functions to disrupt the binding of HilD with promoters, thereby suppressing the transcription of *hilA* and subsequent activation of T3SS-1.

## Administration of fisetin alleviates pathology in the murine *S.* Typhimurium infection model

Next, we established a murine model to evaluate the potential therapeutic effect of fisetin against *S.* Typhimurium infection. The mortality of *S.* Typhimurium-infected mice ($1 \times 10^7$ CFUs per mouse) was monitored for 9 days. As shown in Fig. 8A, mice in the

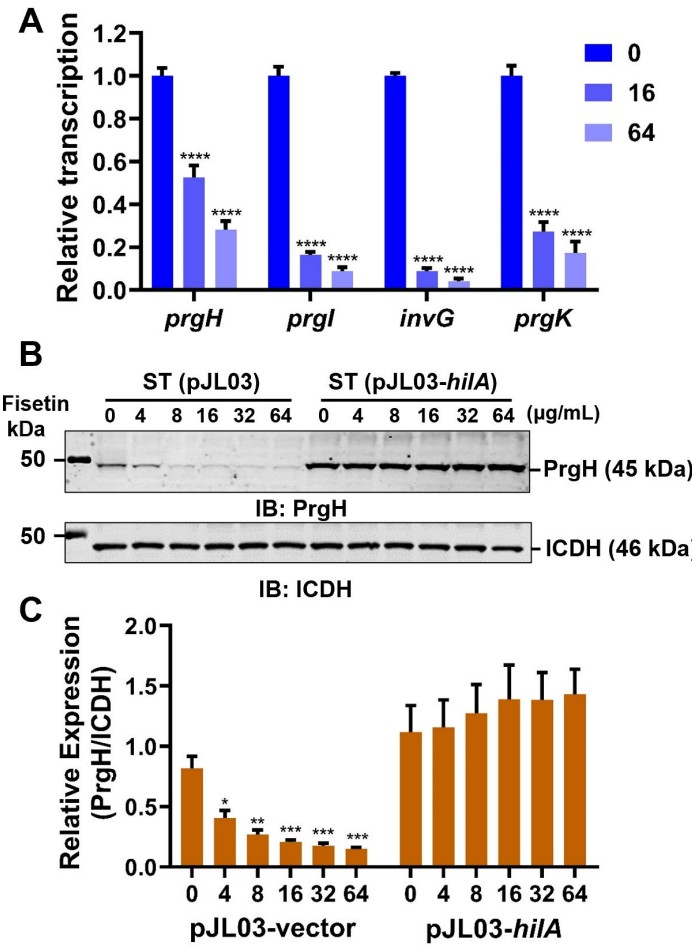

**FIG 6** Fisetin decreases the production of *S*. Typhimurium T3SS-1 structural proteins in a HilA-dependent manner. (A) Transcriptional levels of the T3SS-1 structural genes *prgH*, *prgI*, *invG*, and *prgK* in bacteria receiving fisetin treatment. (B) PrgH protein levels in the bacterial lysates or WT or HilA-overexpressing *S*. Typhimurium strains upon culture with increasing amounts of fisetin. (C) Relative PrgH levels shown in Panel B were evaluated by ImageJ. Data presented in A are the mean ± SD of three independent experiments. ****$P < 0.0001$; ***$P < 0.001$; **$P < 0.01$; *$P < 0.05$.

infection control group all died, while treatment of mice with fisetin showed a certain therapeutic effect, with the survival rate improved to 20%. The organ bacterial burdens in *S*. Typhimurium ($5 \times 10^6$ CFUs per mouse) challenged mice were quantified to further assess the therapeutic activity of fisetin. At 9 days post-infection, the colonization of *S*. Typhimurium in the spleen and liver was decreased by 8.3-fold and 6-fold compared to the infection controls, respectively (Fig. 8B). To further evaluate the influences of fisetin administration on the organ pathological changes in *S*. Typhimurium-infected mice, we performed histopathologic analysis of mouse tissues at 9 days after infection. Gross inspection of the mouse cecum in the infection control group showed less solidified feces, and the terminal tissue of the cecum was shrunken, whereas solidified feces and the cecum were full and flexible in the fisetin-treated mice (Fig. 8C). The pathological tissue section showed that in the infection control mice, hepatocytes were lysed and dissipated, which was accompanied by fatty degeneration and a congested central vein; the spleen was soaked by inflammatory cells, and capillaries showed extensive hyperemia and congestion; the cecum exhibited submucosal edema, and the lumen accumulated a large number of exfoliative necrotic epithelial cells and intestinal villi (Fig. 8D). Compared to the infection control mice, treatment of *S*. Typhimurium-infected mice

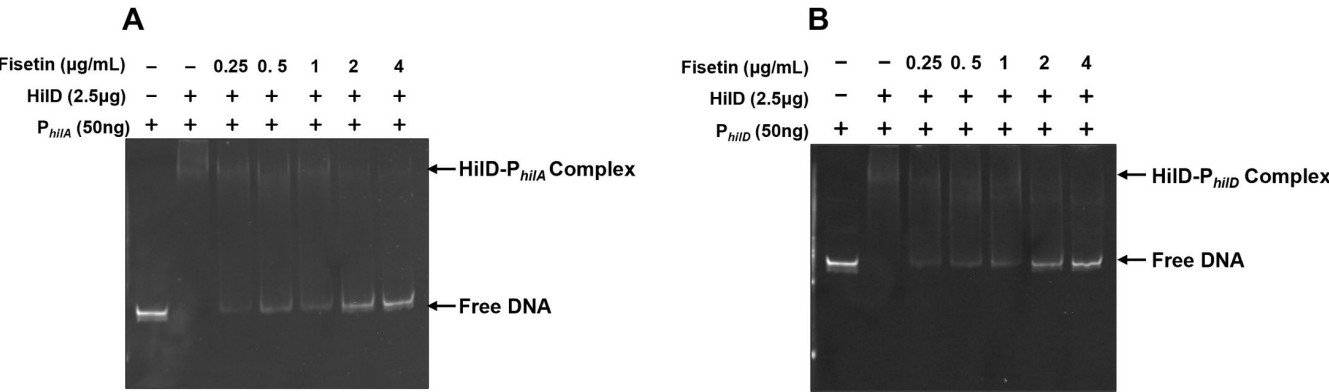

**FIG 7** Fisetin interferes with the binding of HilD with the promoter DNA. Purified recombinant full-length His$_6$-HilD (2.5 µg) was reacted with increasing amounts of fisetin (0–4 µg/mL) and 50 ng of *hilA* (A) or *hilD* (B) promoters for 20 min. The interaction of HilD with relevant promoters was visualized by EMSA. The data shown in A and B are representative of three independent assays.

with fisetin could alleviate the pathological damages of the spleen, liver, and cecum (Fig. 8D).

Finally, we measured the proinflammatory cytokine levels in mouse tissues in both the infection control and fisetin treatment groups. We observed significantly reduced amounts of IL-1β, IL-6, IFN-γ, and TNF-α in the livers of infected mice receiving fisetin. By contrast, all these cytokines in the spleen and cecum were produced at low levels and were not markedly altered after fisetin treatment (Fig. 9A through D).

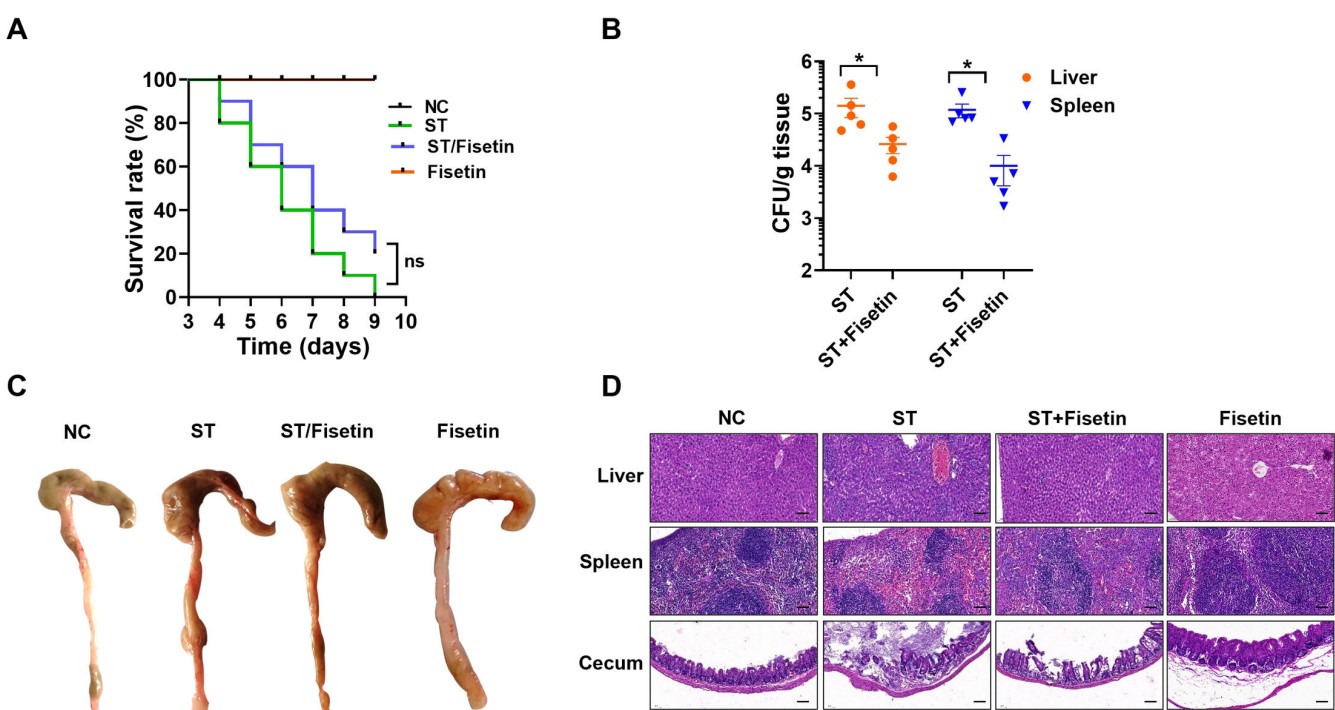

**FIG 8** The therapeutic effects of fisetin treatment on a murine model of *S.* Typhimurium infection. (A) The survival rates of *S.* Typhimurium-infected mice that were either untreated or administered fisetin; each group contained 10 mice ($1 \times 10^7$ CFUs/mouse). (B) The bacterial load in the liver and spleen of infected mice (*n* = 5, $5 \times 10^6$ CFUs/mouse). (C) Gross changes in the cecum in *S.* Typhimurium-infected mice upon treatment with fisetin. (D) Histopathology analysis of the cecum, liver, and spleen in *S.* Typhimurium-infected mice after administration of fisetin. "ST" is an abbreviation of *S.* Typhimurium, "NC" stands for negative control, and "Fisetin" is the fisetin treatment-only group. Data in A and B are representative of three independent tests. Bar, 100 µm. *$P < 0.05$; NS indicates no significant difference.

## DISCUSSION

The traditional antibiotic discovery and development process based on the Waksman platform has not produced any new drugs in the past 30 years. Recently, supported by advances in novel technologies, such as genomics, culturomics, and synthetic chemistry, emerging antibiotic discovery platforms have led to significant progress in the identification of lead compounds to combat AMR (25). For instance, teixobactin produced by *Eleftheria terrae* was first identified by an *in situ* cultivation approach and exhibited excellent antibacterial activity toward gram-positive bacteria, including methicillin-resistant *S. aureus* (MRSA) (22); darobactin, a natural product obtained from screening using *Photorhabdus* symbionts, selectively kills gram-negative bacteria by targeting BamA (26). Apparently, it will be greatly helpful to solve the AMR crisis if these active compounds can be successfully developed and approved for clinical application. However, these strategies will inevitably impose huge selective pressures on the bacteria and result in the emergence of novel bacterial resistance mechanisms owing to their bactericidal or bacteriostatic properties. By contrast, the anti-virulence strategy, which disarms pathogenic bacteria by compromising their critical virulence factors, has gained great interest (27). Since most virulence determinants are not dispensable elements for bacterial growth, anti-virulence drugs are not supposed to impose evolutional pressure as antibiotic treatment (27). In addition, the anti-virulence strategy can preserve beneficial commensal flora due to the lack of direct bacterial killing activity. Indeed, there are examples of anti-virulence drugs that are approved or in development for the treatment of diseases caused by specific pathogenic bacteria (27)

The T3SS is the major virulence factor critical for the pathogenesis of diverse gram-negative bacteria, such as *Salmonella*, *Pseudomonas*, *Shigella*, and *Yersinia* (28, 29). Inactivation of the T3SS by genetic approaches in these bacterial species often leads to loss or alleviation of their pathogenicity. Therefore, the T3SS has been recognized as an attractive target for anti-virulence drug design and development. Importantly, recent striking progress in the understanding of structure, function, and host response to the bacterial T3SS has paved the way for us to rationally screen for active inhibitors.

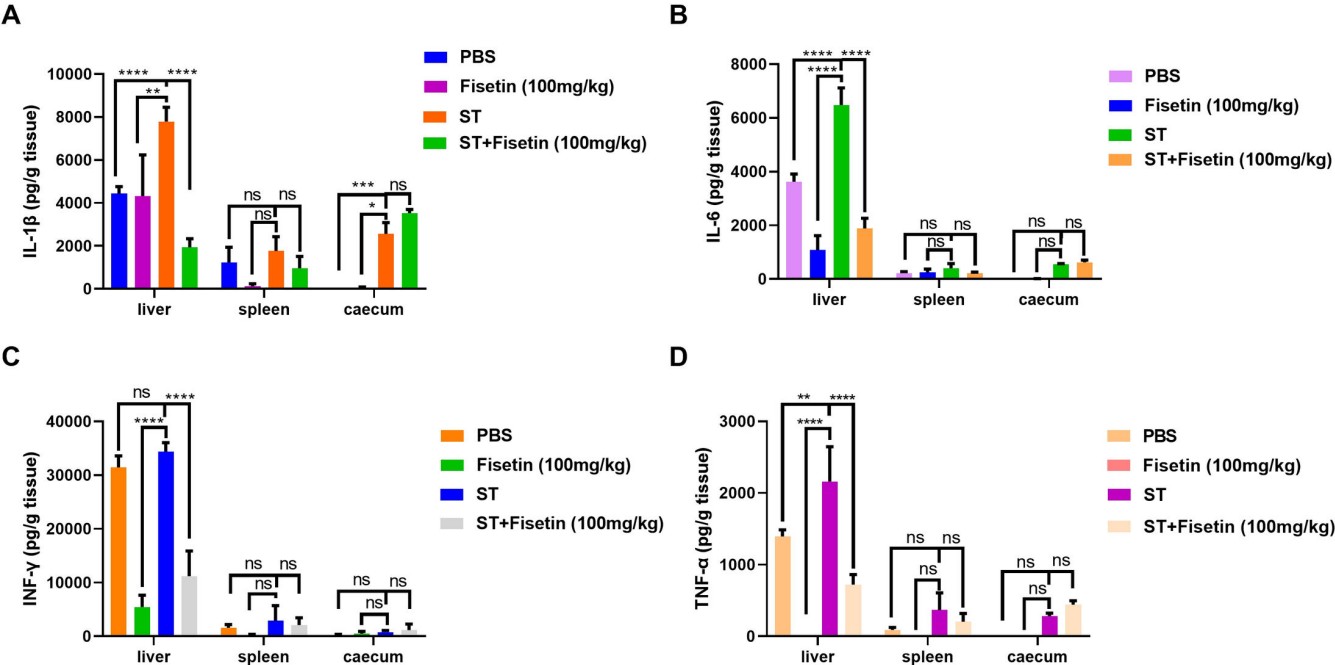

**FIG 9** Fisetin inhibits the production of proinflammatory cytokines in mouse tissues. The cecum, liver, and spleen of *S.* Typhimurium-infected mice were collected and homogenized. The levels of the proinflammatory cytokines IL-1β (A), IL-6 (B), IFN-γ (C), and TNF-α (D) were determined by ELISA. Each group included three mice. Data are representative of three independent experiments. ****$P < 0.0001$; ***$P < 0.001$; **$P < 0.01$; *$P < 0.05$.

Impairment of T3SS can be achieved by targeting the regulation of T3SS expression, the basal body, ATPase activity, the needle complex, and the translocon apparatus *via* small compounds or antibodies (30, 31). The intracellular pathogen *S.* Typhimurium possesses two T3SSs (T3SS-1 and T3SS-2) encoded by individual genomic loci, which play critical roles in bacterial invasion of host cells and intracellular proliferation, respectively (8). In this study, from an invasion-based screening, we demonstrated that fisetin treatment of *S.* Typhimurium was capable of efficiently suppressing its invasion of nonphagocytic cells. In addition, our mechanistic studies have revealed that fisetin could interfere with the interaction of the transcriptional regulator HilD with its relevant promoters, thereby decreasing the transcription of HilA, which further downregulates the expression of T3SS-1 structure proteins and effector proteins. In the murine *S.* Typhimurium infection model, administration of fisetin showed certain therapeutic effects regarding the reduction in bacterial colonization in the organs, alleviation of histopathologic damage, and decrease in proinflammatory cytokine levels in the liver. Therefore, our study indicates that fisetin might potentially be used as a lead compound for anti-virulence drug development targeting T3SS-1. In addition, fisetin might be applied in combination with existing therapies (e.g., antibiotics) to treat infections caused by *Salmonella*. Importantly, our previous research showed that fisetin could inhibit *Listeria monocytogenes* virulence by disrupting the oligomerization of listeriolysin O, and fisetin treatment effectively protected against *L. monocytogenes* infection in the murine infection model (32). Thus, the present study further extends the anti-virulence spectrum of fisetin against distinct bacterial species.

Unlike previously reported T3SS-1 inhibitors, such as harmine (33), tannic acid (34), and hyperoside (35), which can significantly improve the survival rates of *S.* Typhimurium-infected mice, oral administration of *S.* Typhimurium-challenged mice with fisetin exhibits poor survival phenotype. This might be attributable to its low water solubility (36), low bioavailability (37), and poor stability in the intestinal tract (38). Previous animal studies have demonstrated that fisetin undergoes rapid metabolism, enzymatic degradation as well as P-glycoproteins-mediated efflux in the gastrointestinal tract after oral administration (39). These barriers can be overcome by novel formulation technologies such as nanoparticles (39) and hybrid-hydrogel (38). Therefore, it is promising that novel fisetin formulations may improve the survival outcome of *S.* Typhimurium infection.

## MATERIALS AND METHODS

### Bacterial strains, plasmid construction, cell line, culture conditions, and natural compounds

The bacterial strains used in this study are listed in Table S1. All the bacterial species were maintained in Luria-Bertani (LB) agar plates and grown in LB broth. When needed, antibiotics were added to growth media at the following concentrations: streptomycin, 40 µg/mL; gentamicin, 20 µg/mL; kanamycin, 50 µg/mL, and ampicillin, 100 µg/mL. To induce T3SS-1 gene expression, *Salmonella* strains were cultured at 37°C with high-salt LB medium containing 0.3 M NaCl.

For expression of $His_6$-tagged proteins, PCR products using the primers listed in Table S2 were inserted into pET28a and transformed into BL21 (DE3). To overexpress HilA in S. Typhimurium, PCR products of the *hilA* gene were cloned into pJL03 (40), which carried an arabinose promoter, and transformed into relevant strains.

HeLa cells were obtained from the American Type Culture Collection (ATCC). Cells were cultured in Dulbecco's modified Eagle's medium (DMEM) (HyClone) supplemented with 10% fetal bovine serum (FBS) in a humidified $CO_2$ incubator.

All the natural compounds (Purity >98%) were purchased from Chengdu Herbpurify Co., Ltd., and a stock solution was made by dissolving the compounds in DMSO.

## Determination of the minimal inhibitory concentration

The MICs of fisetin against *Salmonella* strains were determined using the broth microdilution method according to the following protocol. In 96-well plates, serial twofold dilutions of fisetin (2,048 to 4 µg/mL) were added to each well. Meanwhile, overnight *Salmonella* cultures were diluted with LB broth containing 0.02% resazurin (wt/vol) to an optical density (OD) of 0.1 at $OD_{600}$ nm, and 100 µL of the diluted culture was added to the 96-well plate. Then, the plates were grown in an incubator at 37°C without agitation for 18 h. The lowest concentration of fisetin that showed a blue change was defined as the MIC of fisetin for *Salmonella*.

## *In vitro* growth curve

Overnight cultures of *Salmonella* were diluted in 200 mL LB at 1:100 and aliquoted into seven sterile Erlenmeyer flasks. Then, fisetin was added to individual aliquots at the indicated concentrations. Cultures were grown at 37°C with consistent agitation at 220 rpm/min. Bacterial growth was measured at 30-min intervals by an ultraviolet spectrophotometer (Biophotometer, Eppendorf). *Salmonella* cultures containing 4 µg/mL kanamycin served as positive controls.

## Bacterial invasion assays

The *Salmonella* culture was diluted in 3 mL 0.3 M NaCl LB at 1:20. Then, increasing concentrations of fisetin (0, 4, 8, 16, 32, and 64 µg/mL) were added to the aliquoted cultures and grown at 37°C with consistent agitation at 220 rpm/min for 4 h. In a 24-well plate, $4 \times 10^5$ HeLa cells were challenged with *Salmonella* strains at an MOI of 100. After infection for 1 h, the cells were washed with prewarmed phosphate-buffered saline (PBS) three times. The plates were replaced with fresh medium containing 100 mg/mL gentamicin and incubated in the incubator for 40 min to kill extracellular bacteria. Cells were washed with PBS and lysed with 0.2% saponin. The lysates were spread on LB plates and grown at 37°C for 18 h before enumerating the CFUs.

To visualize *Salmonella* invasion, HeLa cells were seeded onto coverslips in 24-well plates. Fisetin treatment of *Salmonella* cultures and subsequent infections was performed as described above. Cells were fixed with 4% paraformaldehyde for 20 min at room temperature (RT) followed by permeabilization with 0.02% (vol/vol) Triton X-100 for 20 min. The cells were then blocked with 4% goat serum in PBS for 30 min. *Salmonella* was stained with anti-*Salmonella* antibody for 1 h and washed with PBS three times. Goat anti-rabbit secondary antibody conjugated with Texas Red was used at a 1:500 dilution and stained for 1 h. The nuclei were labeled with Hoechst 33342 diluted in PBS for 5 min. Staining signals were observed under a fluorescence microscope (Olympus IX-83).

## Detection of secreted proteins in *Salmonella* culture supernatant

Overnight *Salmonella* cultures were diluted 1:50 in LB broth containing 0.3 M NaCl. Bacteria were further grown for 4 h at 37°C/220 rpm with or without the addition of fisetin at the indicated concentrations. Culture supernatants were collected by centrifugation at $14,000 \times g$ for 20 min. Then, the supernatants were subjected to trichloroacetic acid (TCA) (10%) precipitation at 4°C for 4 h. After pelleting the secreted proteins by centrifugation at $14,000 \times g$ for 30 min, the precipitates were washed twice with ice-chilled acetone and resuspended in $1 \times$ SDS loading buffer followed by boiling for 5 min. Protein samples were separated by SDS-PAGE and visualized by Coomassie brilliant blue (CBB) staining.

## Recombinant protein purification

pET28a carrying relevant genes was introduced into *Escherichia coli* BL21 (DE3) by transformation. Overnight cultures were diluted into 1 L of fresh LB broth with the addition of 50 µg/mL kanamycin and grown at 37°C/220 rpm until the $OD_{600}$ nm

reached approximately 0.6–0.8. Then, 0.2 mM IPTG was added to the culture to induce the expression of recombinant proteins. Following induction at 18°C for 14 h, bacteria were harvested by centrifugation at 14,000 ×g for 20 min and resuspended in lysis buffer. Then, bacteria were lysed twice by a JN-Mini Low Temperature Ultrahigh Pressure Continuous Flow Cell Cracker (JN-mini, JNBIO, Guangzhou, China). Unbroken cells and debris were removed by centrifugation at 14,000 × g for 1 h. The supernatants were collected and incubated with $Ni^{2+}$-NTA beads for 1 h at 4°C to enrich the recombinant proteins. After washing the beads with buffer containing 50 mM $NaH_2PO_4$, 300 mM NaCl, and 20 mM imidazole, pH 8.0, the bead-bound proteins were eluted by elution buffer containing 50 mM $NaH_2PO_4$, 300 mM NaCl, and 250 mM imidazole, pH 8.0. The resulting protein samples were further dialyzed twice at 4°C in a buffer containing 25 mM Tris-HCl (pH 7.5), 150 mM NaCl, and 10% glycerol.

## Western blot analysis

Polyclonal antibodies specific for SipA, HilA, and PrgH were generated by immunization of rabbits with recombinant proteins following a standard protocol (AbMax Biotechnology Co., Ltd., Beijing, China). *Salmonella* cultures treated with or without fisetin were centrifuged at 12,000 × g for 5 min. Then, the pellets were resuspended in 1× SDS-PAGE sample buffer and boiled for 5 min. The Western blot procedure was performed as previously described (41). The antibodies and their dilutions used in this study were rabbit anti-ICDH (Sigma, cat# ABS2090; 1: 20000); mouse anti-FLAG antibody (Sigma, cat# F1804; 1: 3000); rabbit anti-SipA (1: 1000); rabbit anti-HilA (1: 1000); and rabbit anti-PrgH (1: 1000). The blot signals were observed under the Odyssey CLx Imaging System (Li-Cor).

## β-lactamase reporter assay

*S.* Typhimurium SL1344 (pSipA-TEM) was grown in LB in the presence of gradient concentrations of fisetin to the exponential phase. The bacteria were used to infect HeLa cells that were seeded in 96-well plates at an MOI of 20. Two hours after infection, the cells were washed twice with Hanks' balanced salt solution (HBSS). A 6 × CCF4/AM reaction mixture (K1095; Thermo Fisher) was added to the plates and incubated in the dark at RT for 1 h. SipA-TEM translocation by *S.* Typhimurium was visualized by a fluorescence microscope (IX83; Olympus) equipped with a β-lactamase FL-Cube (U-N41031; Chroma Technology Corp., Bellows Falls, VT).

## Quantitative real-time PCR

*S.* Typhimurium SL1344 was grown in an LB medium containing 0.3 M NaCl with the addition of fisetin. Then, bacterial cells were collected by centrifugation. Total RNA was extracted by a bacterial total RNA extraction kit (B518625; Sangon Biotech) and reverse transcribed into cDNA using a RevertAid RT reverse transcription kit (K1691; Thermo Scientific). The resulting individual cDNA was used as a template for qRT-PCR using SYBR green fluorescent dye (KTSM1401; AlpaLife). The qRT-PCR primers specific for *sipA*, *sipB*, *sipC*, *hilA*, *hilC, hilD, rtsA,* and *gyrB* were described earlier (42); while primers specific for *prgH*, *prgI*, *prgK,* and *invG* are listed in Table S3. The transcription of gyrase subunit B (gyrB) was included as an internal control.

## Electrophoretic mobility shift assay

The EMSA was performed as previously described (43) with slight modifications. The *hilA* and *hilD* promoter regions were amplified from the genomic DNA of *S.* Typhimurium SL1344 using the primers listed in Table S4. Fifty nanograms of the DNA probes, 2.5 µg recombinant $His_6$-HilD, and varying amounts of fisetin were incubated in 20 µL of binding buffer (20 mM KCl, 1 mM DTT, 0.04 mM EDTA, 0.01% Triton X-100, 1% glycerol, 20 mM HEPES, pH 7.2). Reactions were allowed to proceed for 30 min at RT. Then, NativePAGE™ Sample Buffer (4×) was added to stop the reaction, and samples were analyzed with 6% native polyacrylamide gel using TBE running buffer (0.5×). DNA was

stained using SYBR Safe DNA gel stain (Invitrogen) and visualized by a versatile gel image analysis system (Tanon MINI Space, China).

## Animal experiments

Female BALB/c mice (6–8 weeks old, 18–20 g) obtained from Liaoning Changsheng Biotechnology Co., Ltd. were used for all animal experiments in this work. Mice were fed water containing streptomycin (5 g/L) for 3 days prior to *S*. Typhimurium infection. Mice were then randomly divided into the following groups: the control group (without infection), the infection control group (ST group), the fisetin group (without infection), and the fisetin treatment group. To monitor the survival rates, *S*. Typhimurium SL1344 was orally gavaged at $1 \times 10^7$ CFUs in 100 µL PBS (*n* = 10), while for the determination of bacterial colonization in the organs, histopathology observations, and cytokine measurement, mice were orally inoculated with a sublethal dose of *S*. Typhimurium ($5 \times 10^6$ CFUs in 100 µL PBS, *n* = 5). Mice in the fisetin treatment group were orally administered 100 mg/kg fisetin dissolved in a solvent containing 10% DMSO, 40% PEG 400, 45% saline, and 5% Tween-80 at 12-h intervals for the desired durations, while mice in the control group and ST group were administered an equivalent volume of the solvent by the same schedule. Sublethal dose of *S*. Typhimurium-infected mice was euthanized by inhalation of excess carbon dioxide ($CO_2$) at 4 days post-fisetin treatment, and the liver, spleen, and cecum were collected. Then, these tissues were homogenized in PBS and plated on selective LB agar plates containing 20 µg/mL streptomycin for CFU determination. Cytokines in the tissue suspension were measured using ELISA kits (BioLegend) according to the manufacturer's instructions. The liver, spleen, and cecum were fixed in 4% paraformaldehyde, and the fixed tissues were subjected to hematoxylin-eosin (HE) staining.

## Statistical analysis

Data were analyzed using GraphPad Prism 8.0 h (GraphPad Software, La Jolla, CA, USA). The data from the mouse survival assay were assessed using the log-rank test, while the others were analyzed by unpaired two-tailed *t*-tests. The *P* values are shown as follows: ****$P < 0.0001$; ***$P < 0.001$; **$P < 0.01$; *$P < 0.05$.

## ACKNOWLEDGMENTS

This research was supported by the National Key Research & Development Program of China (2021YFD1801000), the Natural Science Foundation of Jilin Province (20230101142JC), The Fundamental Research Funds for the Central Universities, and The Thousand Young Talents Program of the Chinese Government (J.Q.), and startup funds from Jilin University.

## AUTHOR AFFILIATIONS

[1]State Key Laboratory for Diagnosis and Treatment of Severe Zoonotic Infectious Diseases, Key Laboratory for Zoonosis Research of the Ministry of Education, College of Veterinary Medicine, Jilin University, Center for Pathogen Biology and Infectious Diseases, The First Hospital of Jilin University , Changchun, Jilin, China
[2]Key Laboratory for Molecular Enzymology and Engineering of Ministry of Education, School of Life Sciences, Jilin University, Changchun, China
[3]Jiangsu Co-innovation Center for Prevention and Control of Important Animal Infectious Diseases and Zoonoses, Institute of Comparative Medicine, College of Veterinary Medicine, Yangzhou University, Yangzhou, China

## AUTHOR ORCIDs

Hongtao Liu http://orcid.org/0009-0009-2639-2114
Yuan Liu https://orcid.org/0000-0002-9622-6471

Jianfeng Wang  http://orcid.org/0000-0001-8311-0894
Yong Zhang  http://orcid.org/0000-0002-2190-5867
Zhimin Guo  http://orcid.org/0000-0002-2800-5589
Jiazhang Qiu  http://orcid.org/0000-0002-7723-5073

## FUNDING

| Funder | Grant(s) | Author(s) |
|---|---|---|
| National Key Research and Development Program of China | 2021YFD1801000 | Jiazhang Qiu |
| Natural Science Foundation of Jilin Province | 20230101142JC | Jiazhang Qiu |

## AUTHOR CONTRIBUTIONS

Siqi Li, Investigation, Methodology, Validation | Hongtao Liu, Investigation, Supervision | Jingyan Shu, Investigation | Quanshun Li, Data curation, Methodology, Visualization | Yuan Liu, Investigation, Methodology, Supervision | Haihua Feng, Methodology, Supervision | Jianfeng Wang, Project administration, Validation | Xuming Deng, Conceptualization, Supervision | Yong Zhang, Supervision, Writing – original draft, Writing – review and editing | Zhimin Guo, Conceptualization, Investigation, Writing – original draft, Writing – review and editing.

## ADDITIONAL FILES

The following material is available online.

### Supplemental Material

**Supplemental figures (Spectrum02406-23-s0001.doc).** Fig. S1 and S2.
**Supplemental tables (Spectrum02406-23-s0002.doc).** Tables S1 to S4.

### Open Peer Review

**PEER REVIEW HISTORY (review-history.pdf).** An accounting of the reviewer comments and feedback.

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
