## [Reviewer comments · Microbiology Spectrum]

Microbiology Spectrum

Fisetin inhibits *Salmonella* Typhimurium type III secretion system regulator Hild and reduces pathology *in vivo*

Siqi Li, Hongtao Liu, Jingyan Shu, Quanshun Li, Yuan Liu, Haihua Feng, Jianfeng Wang, Xuming Deng, Yong Zhang, Zhimin Guo, and Jiazhang Qiu

Corresponding Author(s): Jiazhang Qiu, Jilin University

Review Timeline:

Submission Date:	June 13, 2023
Editorial Decision:	August 2, 2023
Revision Received:	October 11, 2023
Accepted:	November 14, 2023

Editor: Carlos Blondel

Reviewer(s): Disclosure of reviewer identity is with reference to reviewer comments included in decision letter(s). The following individuals involved in review of your submission have agreed to reveal their identity: Yosef Daniel Huberman (Reviewer #1)

Transaction Report:

DOI: <https://doi.org/10.1128/spectrum.02406-23>

August 2, 2023

Prof. Jiazhang Qiu
Jilin University
College of veterinary medicine
5333# Xi'an Road
Changchun, Jilin 130062
China

Re: Spectrum02406-23 (The natural compound fisetin protects Salmonella Typhimurium infection via inhibition of the type III secretion system)

Dear Prof. Jiazhang Qiu:

Link Not Available

Sincerely,

Carlos Blondel

Journals Department
Reviewer comments:

Reviewer #1 (Comments for the Author):

Introduction.

1. Lines 75-78. Information regarding illness and death in human - readers should be referred to the original publication. Citing (#5) is wrong and should be changed.
2. Lines 78-79. The reference is from 2003 so I suggest looking for more recent works before saying "...gastroenteritis remains one of the primary causes...".
3. Lines 80-81. Please separate the information regarding humans and animals.

4. Lines 109-112. Previous works of the authors. The readers should be referred to published works or this information should be added to the current manuscript.

Results.

5. The title of this manuscript describes assays with *S. Typhimurium*. Nevertheless, some assays were performed with *S. Enteritidis*. I recommend including these results in the manuscript alongside *S. Typhimurium* and not as complementary information.

6. Line 167 and Figure 2D. It is unclear how the secretion levels between treatments were compared for the authors to determine "... that 4 µg/mL fisetin resulted in significantly reduced secretion of SipA and SipB...". Please check.

7. Line 220 and Figure 6. Please name the genes in both places in the same order.

8. Line 237. The HilD promoter complex is not available in the manuscript and should be either described or referenced.

9. Lines 246-251. This information should be transferred to Material and Methods. How did the authors decide about the dose of fisetin and the treatment procedures?

10. Line 271 and Figure 8E. How did the authors compare pathological damage in tissues? It is not clear if they claim that "...fisetin remarkably alleviated the pathological damage...".

Discussion

11. The discussion should discuss the results of this investigation. However, much of it should be transferred into the introduction. Please reorganize this whole section.

12. Line 310. Please cite the statement "...antivirulence drugs are not supposed to".

13. Lines 336 - 337. I do not agree with "... administration of fisetin showed a significant therapeutic effect...". See other comments and please stick to the facts.

14. Line 344. Please state that the animals infected with *L. monocytogenes* were mice.

Materials and Methods

15. Lines 352-353. Bacterial strains that come from previous works of the authors should be referred to published works or more information should be added to the current manuscript.

16. Line 359. Please describe briefly the modification of LB medium.

17. General information regarding routine methods like PCR, thermocycling conditions, western blot, etc should be cited and only modification should be stated.

18. Please add information regarding the definition of the PCR primers that were used. Were they designed by the authors or other publications?

19. The murine trial should be described with more information:

a. The number of mice that were used and justified.

b. Why were only female mice included?

c. A fourth group is missing - treatment with fisetin as a control. The authors should explain why it was not included.

d. More information is needed for the infective dose of *S. Typhimurium* - how was it fixed? In the results (lines 253-254), 100% mortality is described in the non-treated mice and only 20% survival for the treated ones. This is a very poor "improvement". Why was not the LD50 tested and used?

e. The volume of inoculation should be stated.

f. Why histological samples were taken from different mice inoculated with a lower dose of *S. Typhimurium* and not from already dead/euthanized mice of the other trial? Furthermore, it is not clear if the histological samples were taken on day 9 post-infection from dead or euthanized mice.

g. Was the administration of fisetin done individually or in the drinking water?

h. The method of euthanization should be described.

i. Treated mice had significantly reduced bacterial colonization, alleviation of histopathological destruction, and decreased proinflammatory cytokine levels. Nevertheless, mice mortality was only slightly reduced and it was not statistically different so it is not correct to say that the "administration of fisetin can significantly protect *Salmonella* infection in a murine model" (line 37). No real protection was observed. Authors should stick to the facts.

20. Figure 1A. There is no mention of the 29 compounds in the text. Please change the Figure or add more information to the manuscript.

21. Figure 1E. Please define if this figure describes concentrations or OD and use the correct terms in the description.

22. Figure 2. Please add "*****" to $p < 0.0001$.

23. Figures 4A and 6C. Please add significance level and description to only one *.

24. Figure 9. Only one level of significance is shown. Please omit the others.

25. Figure 1S. Please add the description of levels of significance

Staff Comments:

Preparing Revision Guidelines

To submit your modified manuscript, log onto the eJP submission site at <https://spectrum.msubmit.net/cgi-bin/main.plex>. Go to

Author Tasks and click the appropriate manuscript title to begin the revision process. The information that you entered when you first submitted the paper will be displayed. Please update the information as necessary. Here are a few examples of required updates that authors must address:

Please return the manuscript within 60 days; if you cannot complete the modification within this time period, please contact me. If you do not wish to modify the manuscript and prefer to submit it to another journal, please notify me of your decision immediately so that the manuscript may be formally withdrawn from consideration by Microbiology Spectrum.

Title: The natural compound fisetin protects against *Salmonella* Typhimurium infection via inhibition of the type III secretion system

Li, et al

Microbiology Spectrum Review

Overall comments

In general, the paper is well considered. The authors state that there is protection without a significant change in mouse outcomes, which should be avoided. However, the reduction in biomarkers and organ invasion is promising and supports their other conclusions, and their *in vitro* data is thorough. The sum of experimental data here is good and demonstrates a likely mechanism through gel-shift assay, protein abundance, and transcript abundance analysis. They then show activity in a murine model. While a change in death rates is not observed, they show a reduction in important biomarkers, such as the level of organ infiltration and cytokine proliferation.

The paper would benefit from condensing of several figures in the results, though I leave it to the editor whether this is necessary given their journal format. The standard set by the journal is a review of the scientific content and methodological rigor, which is sufficient except for some control data in Figures 8 and 9.

For Figure 8 and 9, there is no control showing the effects of fisetin alone, and the non-infected state should be shown for comparison in both figures as a baseline. Given that fisetin is bioactive in several different contexts, including anti-inflammatory, it is important to control for these effects. Figure 9 needs cytokine levels for the non-infected mice and fisetin alone mice.

The manuscript should be reviewed for introductory and methods material in the results, as well as other issues highlighted in the comments below. Lastly, the discussion has too much introductory information, and should be significantly shortened.

Specific comments

Title: Needs revision. Protection is against infection. Authors demonstrate activity against HilD binding of promoter sequence *in vitro*, and consistent observations in other tests, so title should be re-written to reflect this. Something like: Fisetin inhibits *Salmonella* Typhimurium type III secretion system regulator HilD and reduces pathology *in vivo*.

While I agree that the molecule shows promise and a reduction in disease biomarkers and pathological hallmarks, the lack of a significant difference in mouse outcomes means I advise against claiming protection and encourage other descriptors.

Abstract: Sufficient, but needing some revision to better reflect the results and for improved readability.

Line 25-27: Authors should state the antimicrobial resistance has been rising for *Salmonella* specifically, maybe even mention the emergence of more MDR and newer XDR strains.

Line 26: remove “remarkably”

Line 27: remove “highly”, I would reconsider calling this “necessary”. Consider replacing with “sought” or “desirable”, as in “Hence, alternative therapeutic means to aid treatment of invasive *Salmonella* infections.

Line 29: “plant-derived” is superior to “natural”, remove “natural”.

Line 30: Suggestion: “These efforts identified fisetin as...”

Line 33: Replace “could” with “appears to” or something similarly stronger. Evidence in paper supports stronger language.

Line 37: “significantly protect” should be skipped since there is no improvement in mouse survival. Authors can still reference the biomarkers afterwards as signs that fisetin may have some value.

Importance: Needs revision for readability and English. Some minor revision to content.

Line 49: “urgently demanded” - replace with “needed”, “desired”, or something similar.

Line 52: “targeting T3SS-1” - fisetin appears to target SPI-1 regulation, not the T3SS itself.

Line 56: Again, protection is not warranted here, should mention reduction of biomarkers.

Line 57: Rewrite for clarity and grammar.

Introduction: Generally well considered, but should be shortened to remove unnecessary details, such as the breadth of serovars and their cause as they are not needed to understand the paper, or the details of T3SS-2/SPI-2 function.

Line 83: “misuse” instead of “use”

Line 85: Some more comment on the emergence of XDR Typhi may be warranted here.

Line 105: “primarily present” - consider “naturally found”

Line 116: Recurrent use of protection oversells results.

Line 117: Authors should consider that while they did not see a protection from a death endpoint in the mouse model, fisetin or a related agent may be used in combination with other existing therapies.

Results: Some figures need to be combined to reduce the total number of figures, primarily figures 3 to 7, and Figure 8 and 9. The overall consideration of experiments is good, specific comments will be found line by line below.

Figure 1: Did authors have a positive control for the MIC assay where kanamycin/gentamicin prevented color change. May include with supplementary material, not critical for acceptance. Subfigure C: Use "ST + Fisetin" as in Fig. 9. Subfigure E - include "µg/ml".

Line 122: Introductory material, remove.

Line 140: change to "non-antibacterial fisetin inhibits *S. Typhimurium* invasion."

Figure 2: Consider labeling C and D as "supernatant" to distinguish visually from Figure 3. NOTE: Please include expected molecular weights in figures (MULTIPLE FIGURES) or text.

Line 160: Methods material here isn't needed - curious readers can find this in the methods.

Figure 3: OK

Figure 4: Line 695 - move this to the section on subfigure B at line 688.

Line 191: Introductory material - move to the introduction.

Figure 5: OK

Figure 6: OK

Figure 7: Mislabeled Figure A as "HiID" in image, should be "HiIA".

Figure 8: 8B y-axis label can be simply "CFU/g tissue". Samples should be "ST + Fisetin". Declare that ST is an abbreviation of *S. Typhimurium* in the legend. NC stands for "Negative control" Please specify in legend. Also, include infecting dose, which is different between A and B.

As mentioned above, the experiments need an intervention-only control or the citation or relevant literature.

Line 245: Again, the use of protection here without a mouse endpoint is not warranted.

Line 249: It would help the reader to include a dosing method here, I assume oral gavage?

Line 252: There is no statistical power here to claim a difference.

Line 252: Infectious doses need to be listed to distinguish between 8A and 8B. There is one for determining a survival phenotype, and one that is sub-lethal.

Figure 9: Label y-axes with the cytokine measured. Note use of ST + Fisetin here should be used for Figure 7 and elsewhere. Consider combining with Figure 8. Please include cytokine levels for the control mice as well.

Discussion: This section could use significant revision, and I recommend moving or removing the first two paragraphs. Of note, the XDR/MDR content here should be in the introduction and helps inform the reader.

What is needed here is a broader view of other drugs or treatments that have shown promise or are in development, how the mouse model signs here show promise even though there was no significant survival phenotype. It may also be worth postulating how a drug with activity like that demonstrated in the mouse model would intersect with modern therapy, and whether the authors see this as being developed further into a drug that alone combats disease, or is part of a regimen.

Methods: Appear appropriate and reproducible with the following exception: Multiple manufacturers for different components needed, including for fisetin, and whether a different grade or manufacturer was used between cell culture and mouse experiments.

Supplemental material:

Table S4: The whole sequences probed for the EMSA should be declared along with the promoters used to amplify.

Author's responses to comments

Responses to Reviewer #1

Reviewer #1 (Comments for the Author):

Introduction.

1. Lines 75-78. Information regarding illness and death in human - readers should be referred to the original publication. Citing (#5) is wrong and should be changed.

Response: Thanks for your advice. We have replaced the reference #8.

Line 82: "...million *Salmonella* infection-related annual deaths⁸"

2. Lines 78-79. The reference is from 2003 so I suggest looking for more recent works before saying "...gastroenteritis remains one of the primary causes...".

Response: Thanks for your comment. We have replaced the reference #9 with a recent publication. Please find this in lines 82-84 as well as the reference list in the revised manuscript.

Lines 82-84 in the revised manuscript: "In particular, gastroenteritis remains one of the primary causes of morbidity and mortality in children younger than 5 years⁹."

3. Lines 80-81. Please separate the information regarding humans and animals.

Response: Thanks for your comment. We have revised this sentence.

Lines 87 in the revised manuscript: "...for diseases caused by *Salmonella* in humans and animals¹¹."

4. Lines 109-112. Previous works of the authors. The readers should be referred to published works or this information should be added to the current manuscript.

Response: Thanks for your comment. The screening of invasion inhibitors was conducted in the present study but not from our previous work, and the screening method was shown in the Materials and Methods (Please see the lines 349) and Figure 1A.

Revised Figure 1

Results.

5. The title of this manuscript describes assays with *S. Typhimurium*. Nevertheless, some assays were performed with *S. Enteritidis*. I recommend including these results in the manuscript alongside *S. Typhimurium* and not as complementary information.

Response: Thanks for your comment. We tried to follow your recommendation and combine the results of *S. Enteritidis* and *S. Typhimurium* (Figure 1 plus Supplementary Figure 1). We feel that the combined figure is too repetitious. As we performed the same experiments for *S. Enteritidis*, the data presentation is quite similar as for the *S. Typhimurium*. Therefore, we leave the *S. Enteritidis* in the supplementary figure 1. However, if the reviewer hopes to move this data to the main manuscript, we can present the data as follows.

6. Line 167 and Figure 2D. It is unclear how the secretion levels between treatments were compared for the authors to determine "... that 4 $\mu\text{g/mL}$ fisetin resulted in significantly reduced secretion of SipA and SipB...". Please check.

Response: Thanks for your comment. We have used SipA- and Flag-specific antibodies to detect the secretion levels of SipA and SipB. FliC, the flagellin of bacteria, whose secretion is SPI-1 independent, was used as a loading control. Obviously, compared to the fisetin-free culture, the secretion of SipA and SipB in *S. Typhimurium* cultured with 4 $\mu\text{g/mL}$ fisetin was reduced. To further quantify the alteration between different treatments, we calculated the density ratios of SipA/SipB against FliC using imageJ. Please find these data in revised Figure 2.

Revised Figure 2

7. Line 220 and Figure 6. Please name the genes in both places in the same order.

Response: Thanks for your advice. We have adjusted the order of genes in the manuscript to match those in the Figure 6. Please see the figure 6 and lines 220-222 in the revised manuscript.

Lines 220-222 in the revised manuscript: "...transcripts of *prgH*, *prgI*, *invG*, and *prgK* in *S. Typhimurium* challenged with 64 $\mu\text{g/mL}$ fisetin were decreased by 2.6-, 10.1-, 24-, and 4.9-fold..."

Revised Figure 6

8. Line 237. The HilD promoter complex is not available in the manuscript and should be either described or referenced.

Response: Thanks for your comment. We have revised the labeling of Figure 7. HilD-DNA was changed into HilD- P_{hilA} and HilD- P_{hilD} complex.

Revised Figure 7

9. Lines 246-251. This information should be transferred to Material and Methods. How did the authors decide about the dose of fisetin and the treatment procedures?

Response: Thanks for your suggestion. We have deleted this information as your suggestion.

We have deleted Lines 247-251 in the original submission: “Three days before *S. Typhimurium* infection, mice were administered streptomycin in their drinking water. Mice in the treatment group were given 100 mg/kg fisetin at 12 h intervals, while the control mice were administered an equal volume of solvent on the same schedule.”

In our preliminary experiments, orally administration of mice with 100 mg/kg of fisetin at 12 h-interval for 4 days did not cause toxicity to mice. Due to the long survival time of mice in the *Salmonella* gastroenteritis model, we chose to administer gavage every 12 hours for 4 days in an attempt to observe better treatment outcomes. However, this regimen was still not ideal as no significant improvement of the survival rate was observed. We have discussed the potential reasons in the discussion section.

10. Line 271 and Figure 8E. How did the authors compare pathological damage in tissues? It is not clear if they claim that "...fisetin remarkably alleviated the pathological damage...".

Response: Thanks for your comment. The pathological damages in the tissues were described in Lines 262-268: “The pathological tissue section showed that in the infection control mice, hepatocytes were lysed and dissipated, which was accompanied by fatty degeneration and a congested central vein; the spleen was soaked by inflammatory cells, and capillaries showed extensive hyperemia and congestion; the cecum exhibited submucosal edema, and the lumen accumulated a large number of exfoliative necrotic epithelial cells and intestinal villi (**Fig. 8D**).” In *S. Typhimurium*-infected mice that receiving fisetin treatment, the abovementioned histopathological changes were

alleviated. However, we realized that use “remarkably” is not appropriate here since no quantification was applied. We have revised the description as follows.

Lines 268-270 in the revised manuscript: “Compared to the infection control mice, treatment of *S. Typhimurium*-infected mice with fisetin could alleviate the pathological damages of the spleen, liver, and cecum (Fig. 8D).”

Discussion

11. The discussion should discuss the results of this investigation. However, much of it should be transferred into the introduction. Please reorganize this whole section.

Response: Thanks for your comment. We have reorganized this section. The first paragraph was transferred into the introduction section. In addition, we also discussed the poor survival outcomes of fisetin treatment.

12. Line 310. Please cite the statement "...antivirulence drugs are not supposed to".

Response: Thanks for your suggestion. We have added a reference for this statement. Please find the citation in the reference list (#28)

Line 298: "...supposed to impose evolutionary pressure as antibiotic treatment 28.”

13. Lines 336 - 337. I do not agree with "... administration of fisetin showed a significant therapeutic effect...". See other comments and please stick to the facts.

Response: Thanks for your comment. We have revised the sentence as your suggestion.

Line 324 in the revised manuscript: "... administration of fisetin showed certain therapeutic effect..." the.

14. Line 344. Please state that the animals infected with *L. monocytogenes* were mice.

Response: Thanks for your advice. We have revised it accordingly.

Line 333 in the revised manuscript: "...fisetin treatment effectively protected against *L. monocytogenes* infection in the murine infection model³³."

Materials and Methods

15. Lines 352-353. Bacterial strains that come from previous works of the authors should be referred to published works or more information should be added to the current manuscript.

Response: Thanks for your comment. We have referenced the strains obtained from Dr. Xiaoyun Liu. (Cheng et al., 2017)

Reference #45:

Cheng, S., Wang, L., Liu, Q., Qi, L., Yu, K., Wang, Z., . . . Liu, X. (2017). Identification of a Novel *Salmonella* Type III Effector by Quantitative Secretome Profiling. *Molecular & Cellular Proteomics: MCP*, 16(12), 2219-2228. doi:10.1074/mcp.RA117.000230

16. Line 359. Please describe briefly the modification of LB medium.

Response: Thanks for your comment. In order to stimulate T3SS-1 secretion, *Salmonella* strains were cultured at 37 °C with high-salt LB medium containing 0.3 M NaCl.

Line 360 in the revised manuscript: "*Salmonella* strains were cultured at 37 °C with high-salt LB medium containing 0.3 M NaCl"

17. General information regarding routine methods like PCR, thermocycling conditions, western blot, etc should be cited and only modification should be stated.

Response: Thanks for your advice. We have deleted the description of routine methods accordingly. These procedures were performed according to previous publications.

We have deleted Lines 445-449 in the original submission: "After separation by SDS-PAGE, proteins on the gel were transferred onto PVDF membranes (Pall Life Sciences) by a wet transfer system. Following a blocking step with 5% (w/v) skim milk for 1 h at room temperature, the membranes were probed with appropriate primary antibodies for 1 h."

Lines 449 in the revised manuscript: “The Western blot procedure was performed as previously described⁴².”

18. Please add information regarding the definition of the PCR primers that were used. Were they designed by the authors or other publications?

Response: Thanks for your comment. The qRT-PCR primers that were described in previous studies were deleted and replaced with a reference. Please find the revision in revised Table S2. Other primers listed in Table S2, S3 and S4 were designed in this study.

Lines 472-474 in the revised manuscript: “The qRT-PCR primers specific for *sipA*, *sipB*, *sipC*, *hilA*, *hilC*, *hilD*, *rtsA* and *gyrB* were described earlier⁴³; while primers specific for *prgH*, *prgI*, *prgK* and *invG* were listed in Table S3.”

19. The murine trial should be described with more information:

a. The number of mice that were used and justified.

Response: Thanks for your comment. In the mice survival rates, 10 mice were used in each group. In the experiments involved in inoculating mice with sub-lethal doses, such as bacterial colonization in the organs, histopathology observations, and cytokine measurement, there were five mice each group, where the experiment to detect proinflammatory cytokines used data from only three mice. We have revised in the manuscript.

Lines 498 and 501 in the revised manuscript: “... was orally gavaged at 1×10^7 CFUs in 100 μ L PBS (n=10)” and “... mice were orally inoculated with a sublethal dose of *S. Typhimurium* (5×10^6 CFUs in 100 μ L PBS, n=5).”

b. Why were only female mice included?

Responses: In our preliminary experiments, the gender of the mice did not affect the establishment of the *Salmonella* intestinal infection model. Female mice are more docile than male mice and are more commonly used in *Salmonella* intestinal infections (Gupta, Sehgal, Kanwar, Punj, & Kanwar, 2015; Kong et al., 2008).

References

Gupta, I., Sehgal, R., Kanwar, R. K., Punj, V., & Kanwar, J. R. (2015).

Nanocapsules loaded with iron-saturated bovine lactoferrin have antimicrobial therapeutic potential and maintain calcium, zinc and iron metabolism. *Nanomedicine (London, England)*, 10(8), 1289-1314. doi:10.2217/nnm.14.209

Kong, W., Wanda, S.-Y., Zhang, X., Bollen, W., Tinge, S. A., Roland, K. L., & Curtiss, R. (2008). Regulated programmed lysis of recombinant *Salmonella* in host tissues to release protective antigens and confer biological containment. *Proceedings of the National Academy of Sciences of the United States of America*, 105(27), 9361-9366. doi:10.1073/pnas.0803801105

c. A fourth group is missing - treatment with fisetin as a control. The authors should explain why it was not included.

Responses: Thanks for the comment. In our experiments, we actually contained the drug treatment control. Since treatment with fisetin could not count the bacterial load of organs, these data were deleted in order to maintain the consistency of Figure 8B with other panels. According to your suggestion, we have added the group of only treatment with fisetin to Figure 8. Please find our revision in revised Figure 8A, Figure 8C and Figure 8D.

Revised Figure 8

d. More information is needed for the infective dose of *S. Typhimurium* - how was it fixed? In the results (lines 253-254), 100% mortality is described in the non-treated mice and only 20% survival for the treated ones. This is a very poor "improvement". Why was not the LD50 tested and used?

Response: Thanks for your comment. The infective doses of *S. Typhimurium* were fixed based on our extensive preliminary experiments. 1×10^7 CFUs is the minimal infection dose that can cause death of all infected mice within 9 days. While 5×10^6 CFUs is the optimal dose that causes significant pathological changes and organ bacterial colonization without leading to any death.

20% survival is indeed poor "improvement". No toxicity was reported up to 2000 mg/kg b. wt. of fisetin when administered orally as a single oral dose to mice (Currais et al., 2014). Therefore, the LD₅₀ should be very high. We actually tried higher doses which did not improve the survival rate either. We have analyzed the potential reasons in the discussion section.

Reference

Currais, A., Prior, M., Dargusch, R., Armando, A., Ehren, J., Schubert, D., . . . Maher, P. (2014). Modulation of p25 and inflammatory pathways by fisetin maintains cognitive function in Alzheimer's disease transgenic mice. *Aging Cell*, 13(2), 379-390. doi:10.1111/accel.12185

Lines 338-347 in the revised manuscript: "oral administration of *S. Typhimurium*-challenged mice with fisetin exhibits poor survival phenotype...Therefore, it is promising that novel fisetin formulations may improve the survival outcome of *S. Typhimurium* infection."

e. The volume of inoculation should be stated.

Response: Thanks for your suggestion. The volume of inoculation was 100 μ L. Lines 498 and 501 in the revised manuscript: "To monitor the survival rates, *S. Typhimurium* SL1344 was orally gavaged at 1×10^7 CFUs in 100 μ L PBS (n=10)" and "...mice were orally inoculated with a sublethal dose of *S. Typhi-*

murium (5×10^6 CFUs in 100 μ L PBS, n=5)”

f. Why histological samples were taken from different mice inoculated with a lower dose of *S. Typhimurium* and not from already dead/euthanized mice of the other trial? Furthermore, it is not clear if the histological samples were taken on day 9 post-infection from dead or euthanized mice.

Response: Thanks for your comment. For the organ bacterial colonization and histological samples, we inoculated mice with sublethal doses of *S. Typhimurium*. We thought that using lethal dose of bacteria might introduce many uncertain and artificial factors. For example, each infected mice may die at different days post-infection. The histological samples from different group were taken on day 4 post-infection from euthanized mice. We have included these descriptions in the Materials and Method section.

Lines 506-508 in the revised manuscript: “Sublethal dose of *S. Typhimurium* infected mice were euthanized by inhalation of excess carbon dioxide (CO₂) at 4 days post fisetin treatment, and the liver, spleen and cecum were collected.”

g. Was the administration of fisetin done individually or in the drinking water?

Response: Thanks for the comment. Infected mice were administrated with fisetin individually.

Lines 502-503 in the revised manuscript: “...orally administered 100 mg/kg fisetin dissolved in a solvent containing 10% DMSO, 40% PEG 400, 45% saline, and 5% Tween-80...”.

h. The method of euthanization should be described.

Response: Thanks for your comment. We have revised our manuscript as your suggestion and described the euthanization method.

Lines 506-507 in the revised manuscript: “Sublethal dose of *S. Typhimurium* infected mice were euthanized by inhalation of excess carbon dioxide (CO₂) ...”

i. Treated mice had significantly reduced bacterial colonization, alleviation of histopathological destruction, and decreased proinflammatory cytokine levels.

Nevertheless, mice mortality was only slightly reduced and it was not statistically different so it is not correct to say that the "administration of fisetin can significantly protect *Salmonella* infection in a murine model" (line 37). No real protection was observed. Authors should stick to the facts.

Response: Thanks. We agree with your comment and have revised our manuscript accordingly.

Lines 38-41 in the revised manuscript: "In addition, administration of fisetin in the *Salmonella* murine infection model resulted in reduced bacterial colonization, alleviation of histopathological destruction, and decreased proinflammatory cytokine levels."

20. Figure 1A. There is no mention of the 29 compounds in the text. Please change the Figure or add more information to the manuscript.

Response: Thanks for your comments. We have revised Figure 1A as your suggestion.

Revised Figure 1

21. Figure 1E. Please define if this figure describes concentrations or OD and use the correct terms in the description.

Response: Thanks for your comments. The initial OD_{600nm} of the growth curve is 0.1, and this value is taken as a log, resulting in a number of -1. Since the stationary growth phase of *Salmonella* is OD_{600nm} around 2.5, the values of

this figure range from -1 to 1. We added the concentration of fisetin to the **Figure 1E** and **Figure S1D**.

Revised Figure 1

Revised Supplementary Figure 1

22. Figure 2. Please add "*****" to $p < 0.0001$.

Response: Thanks for your comment. "***** $P < 0.0001$ " have been added in the manuscript.

"****, $P < 0.0001$, $P < 0.0001$; ***, $P < 0.001$ " Please see the line 708 of the revised manuscript.

23. Figures 4A and 6C. Please add significance level and description to only

one *.

Response: Thanks for your comment. “*, $P < 0.05$.” have been added in the manuscript. Please see the lines 739 and 761 of the revised manuscript.

Lines 739 in the revised manuscript: “... **, $P < 0.01$; *, $P < 0.05$.”

Lines 761 in the revised manuscript: “... $P < 0.001$; **, $P < 0.01$; *, $P < 0.05$.”

24. Figure 9. Only one level of significance is shown. Please omit the others.

Response: Thanks for your comment. Figure 9 added data. The level of significance was added to its legend.

“... ****, $P < 0.0001$; ***, $P < 0.001$; **, $P < 0.01$; *, $P < 0.05$.” Please see the line 794 of the revised manuscript.

25. Figure 1S. Please add the description of levels of significance.

Response: Thanks for your comment. “****, $P < 0.0001$; **, $P < 0.01$ ” have been added in the manuscript. Please see the line 807 of the revised manuscript.

Line 807 in the revised manuscript: “... independent experiments. ****, $P < 0.0001$; **, $P < 0.01$.”

Responses to Reviewer #2

Reviewer #2 (Comments for the Author):

Title: The natural compound fisetin protects against *Salmonella* Typhimurium infection via inhibition of the type III secretion system

Li, et al

Microbiology Spectrum Review

Overall comments

In general, the paper is well considered. The authors state that there is protection without a significant change in mouse outcomes, which should be avoided. However, the reduction in biomarkers and organ invasion is promising and supports their other conclusions, and their *in vitro* data is thorough. The sum of experimental data here is good and demonstrates a likely mecha-

nism through gel-shift assay, protein abundance, and transcript abundance analysis. They then show activity in a murine model. While a change in death rates is not observed, they show a reduction in important biomarkers, such as the level of organ infiltration and cytokine proliferation.

Response: Thanks for your comment.

The paper would benefit from condensing of several figures in the results, though I leave it to the editor whether this is necessary given their journal format. The standard set by the journal is a review of the scientific content and methodological rigor, which is sufficient except for some control data in Figures 8 and 9.

Response: Thanks for your comment. We actually have tried to combine Figure 8 and 9 into one figure. However, it looks too crowd. We went through recent issues of Microbiology spectrum, and 9 main figures is allowed for the journal.

For Figure 8 and 9, there is no control showing the effects of fisetin alone, and the non-infected state should be shown for comparison in both figures as a baseline. Given that fisetin is bioactive in several different contexts, including anti-inflammatory, it is important to control for these effects. Figure 9 needs cytokine levels for the non-infected mice and fisetin alone mice.

Response: Thanks for your comment. In our experiments, we actually contained fisetin alone group. However, since treatment with fisetin could not count the bacterial load of organs, these data were not included in our original submission in order to maintain the consistency of Figure 8B with other experiments. In the revised Figure 8 and 9, we have added these data.

The manuscript should be reviewed for introductory and methods material in the results, as well as other issues highlighted in the comments below. Lastly, the discussion has too much introductory information, and should be significantly shortened.

Response: Thanks for your comment. We have carefully reviewed the results section and removed some introductory and method descriptions. Please find

these in our detailed responses to specific comments. In addition, we also re-organized the discussion section as your suggestions.

Specific comments

1. Title: Needs revision. Protection is against infection. Authors demonstrate activity against HilD binding of promoter sequence *in vitro*, and consistent observations in other tests, so title should be re-written to reflect this. Something like: Fisetin inhibits *Salmonella* Typhimurium type III secretion system regulator HilD and reduces pathology *in vivo*.

While I agree that the molecule shows promise and a reduction in disease biomarkers and pathological hallmarks, the lack of a significant difference in mouse outcomes means I advise against claiming protection and encourage other descriptors.

Response: Thanks for the constructive comments. We have realized that using “protection” is not appropriate according to the poor mouse outcomes in the survival rates experiment. Therefore, we have revised the title as your advice.

The revised title: **“Fisetin inhibits *Salmonella* Typhimurium type III secretion system regulator HilD and reduces pathology *in vivo*”**

2 Abstract: Sufficient, but needing some revision to better reflect the results and for improved readability.

Response: Thanks for your comment. We have made changes for the “abstract” section according to your detailed suggestions.

Line 25-27: Authors should state the antimicrobial resistance has been rising for *Salmonella* specifically, maybe even mention the emergence of more MDR and newer XDR strains.

Response: Lines 26-27 in the revised manuscript: **“...especially the emergence of more MDR and newer XDR strains...”**

Line 26: remove “remarkably”

Response: Line 27 in the revised manuscript, “remarkably” was removed.

We have deleted Line 26 in the original submission: "...has remarkably compromised the efficacy of conventional..."

Line 27 in the revised manuscript: "...has compromised the efficacy of conventional antimicrobial therapy for *Salmonella* infections..."

Line 27: remove "highly", I would reconsider calling this "necessary". Consider replacing with "sought" or "desirable", as in "Hence, alternative therapeutic means to aid treatment of invasive *Salmonella* infections."

Response: Line 29 in the revised manuscript: "highly" was removed and "necessary" was substituted with "desirable".

Line 29 in the revised manuscript: "Hence, it is desirable to develop alternative therapeutic means..."

Line 29: "plant-derived" is superior to "natural", remove "natural".

Response: Line 30 in the revised manuscript, we have used "plant-derived" instead of "natural".

"...we screened plant-derived compounds to identify inhibitors of *Salmonella* invasion..." in line 30.

Line 30: Suggestion: "These efforts identified fisetin as..."

Response: Line 32 in the revised manuscript: "These efforts identified fisetin as..."

Line 33: Replace "could" with "appears to" or something similarly stronger. Evidence in paper supports stronger language.

Response: Line 35 in the revised manuscript: "Fisetin appears to interfere with the interaction between HiID and its..."

Line 37: "significantly protect" should be skipped since there is no improvement in mouse survival. Authors can still reference the biomarkers afterwards as signs that fisetin may have some value.

Response: Thanks for your advice. We have deleted "administration of fisetin can significantly protect *Salmonella* infection in a murine model" and rephrased this sentence.

Lines 38-41 in the revised manuscript: "In addition, administration of fisetin in

the *Salmonella* murine infection model resulted in reduced bacterial colonization, alleviation of histopathological destruction, and decreased proinflammatory cytokine levels.”

3. Importance: Needs revision for readability and English. Some minor revision to content.

Response: Thanks for your comment. We have revised the “Importance section” according to your suggestions. Please find these changes as described below.

Line 49: “urgently demanded” - replace with “needed”, “desired”, or something similar.

Response: Lines 49 in the revised manuscript, we have replaced “urgently demanded” with “needed”.

“...novel anti-infection drugs or strategies are needed.” in line 49.

Line 52: “targeting T3SS-1” - fisetin appears to target SPI-1 regulation, not the T3SS itself.

Response: Lines 52 in the revised manuscript: “...could inhibit *Salmonella* invasion of host cells by targeting SPI-1 regulation.”

Line 56: Again, protection is not warranted here, should mention reduction of biomarkers.

Response: Thanks. We have rephrased this sentence. Lines 55-58 in the revised manuscript: “Moreover, administration of fisetin could reduce pathology in the *Salmonella* murine infection model...compound for the development of anti-*Salmonella* drugs.”

Line 57: Rewrite for clarity and grammar.

Response: Lines 56-58 in the revised manuscript: “Collectively, our results suggest that fisetin may serve as a promising lead compound for the development of anti-*Salmonella* drugs.”

4. Introduction: Generally well considered, but should be shortened to remove unnecessary details, such as the breadth of serovars and their cause as they are not needed to understand the paper, or the details of T3SS-2/SPI-2 func-

tion.

Response: Thanks for your comment. We have removed unnecessary details based on your suggestion.

(1) The XDR/MDR content is added in the introduction and helps the reader understand the antibiotic resistance situations

Lines 65-76 in the revised manuscript: “The increasing antimicrobial resistance (AMR) in diverse bacterial pathogens has raised tremendous...antibiotics or therapeutic strategies to tackle this crisis.”

(2) The breadth of serovars is removed.

We have deleted Lines 64-75 in the original submission: “*Salmonella enterica* is a facultative intracellular bacterial pathogen that is capable of causing localized or systemic infections in both humans and animals...and serovar Enteritidis (*S. Enteritidis*) are representative nontyphoidal *Salmonella* (NTS) that cause diarrheal disease in human and various animal hosts ⁴”

Line 83: “misuse” instead of “use”

Response: Line 88 in the revised manuscript, “use” was replaced with “misuse”.

“Unfortunately, the widespread misuse of antibiotics has greatly...” in line 88.

Line 85: Some more comment on the emergence of XDR Typhi may be warranted here.

Response: Thanks for your comment. We have added some comments of XDR Typhimurium.

Lines 91-92 in the revised manuscript: “In particular, the emergence of extensive drug-resistant (XDR) *Salmonella* strains has raised global public health concerns¹³.”

Line 105: “primarily present” - consider “naturally found”

Response: Line 114 in the revised manuscript, “primarily present” was replaced with “naturally found”.

Line 114 in the revised manuscript: “Fisetin is a flavonoid compound that is naturally found in various...”.

Line 116: Recurrent use of protection oversells results.

Response: Lines 124-127 in the revised manuscript: “In addition, fisetin treatment showed reduced organ bacterial colonization, alleviated histopathological destruction, and decreased cytokine levels in a murine *Salmonella* infection model”.

Line 117: Authors should consider that while they did not see a protection from a death endpoint in the mouse model, fisetin or a related agent may be used in combination with other existing therapies.

Response: Thanks for your comment. We have revised our manuscript as your suggestion.

Lines 129-130 in the revised manuscript: “Moreover, fisetin may be applied in combination with existing therapies (e. g., antibiotics) for the treatment of *Salmonella* infections”.

5. Results: Some figures need to be combined to reduce the total number of figures, primarily figures 3 to 7, and Figure 8 and 9. The overall consideration of experiments is good, specific comments will be found line by line below.

Response: Thanks for your comments.

Figure 1: Did authors have a positive control for the MIC assay where kanamycin/gentamicin prevented color change. May include with supplementary material, not critical for acceptance. Subfigure C: Use “ST + Fisetin” as in Fig. 9. Subfigure E - include “ $\mu\text{g/ml}$ ”.

Response: Thanks for your comments. *S. Typhimurium* SL1344 is a standard strain that has very clear genomic background. SL1344 is sensitive to kanamycin which is showed in the growth curve assay showed in Fig. 1E.

We have followed your suggestions and used “ST + Fisetin” in Figure 1.

And we have revised in Figure 1C. Please see the **Figure 1**.

Revised Figure 1

Line 122: Introductory material, remove.

Response: Thanks. We have deleted Lines 122-124 in the original submission: “Successful invasion of nonphagocytic cells is a critical step in Salmonella pathogenesis, suggesting that the blockage of invasion might be an effective means to control Salmonella infection ⁸”

Line 134-136 in the revised manuscript: “In this study, we screened 550 natural compounds...HeLa cells via the gentamicin protection assay”

Line 140: change to “non-antibacterial fisetin inhibits S. Typhimurium invasion.”

Response: Thanks for your comment.

Lines 150-151 in the revised manuscript: “...non-antibacterial fisetin inhibits *Salmonella* invasion”.

Figure 2: Consider labeling C and D as “supernatant” to distinguish visually from Figure 3. NOTE: Please include expected molecular weights in figures (MULTIPLE FIGURES) or text.

Response: Thanks for your comments. We have revised Figure 2 as your suggestion. In addition, we have added the expected molecular weights in each western blot images.

Revised Figure 2

Line 160: Methods material here isn't needed - curious readers can find this in the methods.

Response: Thanks for your comment. We have deleted Lines 160-163 in the original submission: "S. Typhimurium was cultured in the presence or absence of fisetin for 5 h. Then, the culture supernatants were collected and subjected to TCA precipitation. After SDS-PAGE, the secreted proteins were detected by Coomassie brilliant blue (CBB) staining."

Figure 3: OK

Response: Thanks.

Figure 4: Line 695 - move this to the section on subfigure B at line 688.

Response: Thanks for your comment. We have moved line 695 to the section on subfigure B.

Lines 733-734 in the revised manuscript: "...Data shown in B are the mean \pm SD of three independent experiments. (C) HilA production..."

Line 191: Introductory material - move to the introduction.

Response: Thanks for your comment. We have moved line 191 Introductory material to the introduction.

Lines 104-112 in the revised manuscript: "The expression of T3SS-1 genes is tightly controlled by...and activate the expression of *hilC*, *rtsA* and *hilA*"¹¹

Figure 5: OK

Response: Thanks for your comment.

Figure 6: OK

Response: Thanks for your comment.

Figure 7: Mislabeled Figure A as “HiID” in image, should be “HilA”.

Response: Thanks for your comment. We have checked the Figure carefully and it should be HilD. HilD protein can bind to the *hilA* and *hilD* promoters, thus driving the transcription of HilA and HilD.

Figure 8: 8B y-axis label can be simply “CFU/g tissue”. Samples should be “ST + Fisetin”. Declare that ST is an abbreviation of *S. Typhimurium* in the legend. NC stands for “Negative control” Please specify in legend. Also, include infecting dose, which is different between A and B.

As mentioned above, the experiments need an intervention-only control or the citation or relevant literature.

Response: Thanks for your comment. We have revised Figure 8 as well as the legend according to your suggestions. Moreover, we added data of the fisetin treatment only group. Please find these changes in our revised Figure 8 and lines 778–784 in the revised manuscript.

Revised Figure 8

“...each group contained 10 mice (1×10^7 CFUs/mouse). (B) The bacterial load ...“ST” is an abbreviation of *S. Typhimurium*, “NC” stands for Negative control, “Fisetin” is fisetin treatment only group.” in lines 778–784.

Line 245: Again, the use of protection here without a mouse endpoint is not warranted.

Response: Thanks. We have revised this sentence with “Administration of fisetin alleviates pathology in the murine *S. Typhimurium* infection model”. Please find these changes in lines 245-246 of the revised manuscript.

Line 249: It would help the reader to include a dosing method here, I assume oral gavage?

Response: Thanks for your comment. The dosing method was described in the materials and methods section. While another reviewer suggested that these details should not be presented in the results section, we have deleted these sentences.

We have deleted Lines 247-251 in the original submission: “Three days before *S. Typhimurium* infection, mice were administered streptomycin in their drinking water. Mice in the treatment group were given 100 mg/kg fisetin at 12 h intervals, while the control mice were administered an equal volume of solvent on the same schedule.”

Line 252: There is no statistical power here to claim a difference.

Response: Thanks for your comment. “protective” was replaced with “therapeutic” in the manuscript.

Line 251 in the revised manuscript: “...mice with fisetin showed a certain therapeutic effect...”

Line 252: Infectious doses need to be listed to distinguish between 8A and 8B. There is one for determining a survival phenotype, and one that is sub-lethal.

Response: Thanks for your comment. We have added the infectious doses for both lethal and sub-lethal infection. Please find these information in Lines 249 and 253 of the revised manuscript.

Line 249: “The mortality of *S. Typhimurium*-infected mice (1×10^7 CFUs per mouse) was ...”

Line 253: “The organ bacterial burdens in *S. Typhimurium* (5×10^6 CFUs per mouse) challenged mice...”

Figure 9: Label y-axes with the cytokine measured. Note use of ST + Fisetin here should be used for Figure 7 and elsewhere. Consider combining with Figure 8. Please include cytokine levels for the control mice as well.

Response: Thanks for your comment. We have tried to integrate Figure 8 and 9 into one figure. However, it looks crowd. In addition, we went through recent issues of the journal, 9 main figures are allowed. Therefore, we have kept these figures in the current form. If the editor recommends to condense the figures, we can re-organize them.

We have labeled y-axes of Figure 9 with each cytokine measured. In addition, we have added the data for control mice.

Revised Figure 9

6. Discussion: This section could use significant revision, and I recommend moving or removing the first two paragraphs. Of note, the XDR/MDR content here should be in the introduction and helps inform the reader.

What is needed here is a broader view of other drugs or treatments that have shown promise or are in development, how the mouse model signs here show promise even though there was no significant survival phenotype. It may also be worth postulating how a drug with activity like that demonstrated in the mouse model would intersect with modern therapy, and whether the authors see this as being developed further into a drug that alone combats disease, or

is part of a regimen.

Response: Thanks for your suggestions. We have re-organized the discussion section.

1) The first paragraph was transferred into the introduction which may help the readers to understand the antibiotic resistance situations.

2) We kept the second paragraph in the discussion section. We believed that these information would inform readers the strengths and weaknesses of both antimicrobial drug discovery and the anti-virulence strategy.

3) We have described the potential application of fisetin with modern therapies. Lines 328-330 in the revised manuscript: “In addition, fisetin might be applied in combination with existing therapies (e. g., antibiotics) to treat infections caused by *Salmonella*.”

4) We have discussed the potential reasons for the poor survival phenotypes in the therapeutic study.

Lines 336-347 in the revised manuscript: “Unlike previously reported T3SS-1 inhibitors...Therefore, it is promising that novel fisetin formulations may improve the survival outcome of *S. Typhimurium* infection.”

7. Methods: Appear appropriate and reproducible with the following exception: Multiple manufacturers for different components needed, including for fisetin, and whether a different grade or manufacturer was used between cell culture and mouse experiments.

Response: Thanks for your comment. We have added the manufacturers for some critical reagents as well as the purity of the fisetin. In the cell culture and mouse experiments, fisetin was from the same manufacturer and same grade.

Line 371: “All the natural compounds (Purity > 98%) were purchased from Chengdu Herbpurify Co., Ltd.”

Supplemental material:

8 Table S4: The whole sequences probed for the EMSA should be declared along with the promoters used to amplify.

Response: Thanks for your comment. The whole sequences of the promoters have added in the revised manuscript and included in revised Table S4.

Re: Spectrum02406-23R1 (Fisetin inhibits *Salmonella* Typhimurium type III secretion system regulator HilD and reduces pathology *in vivo*)

Dear Prof. Jiazhang Qiu:

Your manuscript has been accepted, and I am forwarding it to the ASM production staff for publication. Your paper will first be checked to make sure all elements meet the technical requirements. ASM staff will contact you if anything needs to be revised before copyediting and production can begin. Otherwise, you will be notified when your proofs are ready to be viewed.

Sincerely,
Carlos Blondel
Editor
Microbiology Spectrum